

# Automated mapping of the seasonal evolution of surface meltwater and its links to climate on the Amery Ice Shelf, Antarctica

Peter A. Tuckett[1], Jeremy C. Ely[1], Andrew J. Sole[1], James M. Lea[2], Stephen J. Livingstone[1], Julie M. Jones[1], J. Melchior van Wessem[3]

[1]Department of Geography, University of Sheffield, Sheffield, S3 7ND, UK
[2]Department of Geography, University of Liverpool, Liverpool, UK
[3]Institute for Marine and Atmospheric Research, Utrecht University, Utrecht, the Netherlands

Correspondence to: Peter A. Tuckett (patuckett1@sheffield.ac.uk)

**Abstract.** Surface meltwater is widespread around the margin of the Antarctic Ice Sheet and has the potential to influence ice-shelf stability, ice-dynamic processes and ice-albedo feedbacks. Whilst the general spatial distribution of surface meltwater across the Antarctic continent is now relatively well known, our understanding of the seasonal and multi-year evolution of surface meltwater is limited. Attempts to generate robust time series of melt cover have largely been constrained by computational expense or limited ice surface visibility associated with mapping from optical satellite imagery. Here, we implement an existing meltwater detection method alongside a novel method for calculating visibility metrics within Google Earth Engine. This enables us to quantify uncertainty induced by cloud cover and variable image data coverage, allowing us to automatically generate time series of surface melt area over large spatial and temporal scales. We demonstrate our method on the Amery Ice Shelf region of East Antarctica, analysing 4,164 Landsat 7 and 8 optical images between 2005 and 2020. Results show high interannual variability in surface meltwater cover, with mapped cumulative lake area totals ranging from 384 km$^2$ to 3898 km$^2$ per melt season. However, by incorporating image visibility assessments into our results, we estimate that cumulative total lake areas are on average 42 % higher than minimum mapped values, highlighting the importance of accounting for variations in image visibility when mapping lake areas. In a typical melt season, total lake area remains low throughout November and early December, before increasing, on average, by an order of magnitude during the second half of December. Peak lake area most commonly occurs during January, before decreasing during February as lakes freeze over. We show that modelled melt predictions from a regional climate model provides a good indication of lake cover in the Amery region, and that annual lake coverage is strongly associated with phases of the Southern Annular Mode (SAM); surface melt is typically highest in years with a negative austral summer SAM index. Furthermore, we suggest that melt-albedo feedbacks modulate the spatial distribution of meltwater in the region, with the exposure of blue ice from persistent katabatic wind scouring influencing the susceptibility of melt ponding. Results demonstrate how our method could be scaled up to generate a multi-year time series record of surface water extent from optical imagery at a continent-wide scale.



# 1 Introduction

Surface meltwater has been known to exist in Antarctica since the early 20th century, when explorers noted the presence of thaw-water streams on the Nansen Ice Shelf (David & Priestly, 1909). The advent of remote sensing techniques during the latter half of the 20th century enabled the identification of surface streams, lakes and ponds in several regions of Antarctica,

including the Antarctic Peninsula (Scambos et al., 2000) and selected glacier basins in East Antarctica (Phillips, 1998; Kingslake et al., 2015; Langley et al., 2016). Until recently, the occurrence of surface meltwater was considered spatially limited. Kingslake et al. (2017), however, demonstrated that surface meltwater is widespread across the Antarctic continent, and subsequently, we now have a reasonable understanding of the spatial distribution of Antarctic surface meltwater (Stokes et al., 2019; Liang et al., 2021). The majority of surface melting occurs at lower latitudes and elevations of the ice-sheet

periphery (Kingslake et al., 2017), with ponding of surface meltwater particularly abundant on relatively flat ice shelf surfaces (Alley et al., 2018; Stokes et al., 2019). Surface lakes and streams can also form within the ice-sheet grounding zone where katabatic winds, which are concentrated at the break of slope, displace colder and damper air adjacent to the ice surface (Lenearts et al., 2017). Surface snow scouring by katabatic winds can additionally amplify albedo effects associated with blue-ice areas or exposed nunataks, which can promote surface melting at a localised scale (Kingslake et al., 2017; Arthur et al.,

2020a; Jakobs et al., 2021). Although our understanding of what controls the spatial distribution of surface meltwater is increasing, our understanding of surface lake evolution throughout melt seasons and on a multi-year timescale remains limited (Arthur et al., 2020b).

Understanding the evolution of surface meltwater in Antarctica is important as it has the potential to influence ice dynamic processes and ice-albedo feedbacks in several ways (Bell et al., 2018). First, melting at the ice surface can directly lead to mass

loss from ablation and runoff. Whilst this is a major contributor to mass loss from the Greenland ice sheet (Shepherd et al., 2020), the majority of surface melt on grounded ice in Antarctica refreezes in situ, and therefore contributes a negligible amount to mass loss (Smith et al., 2020). Second, meltwater ponding on ice shelves can trigger their catastrophic break-up via processes of shelf flexure and hydrofracture (Scambos et al., 2000; Banwell et al., 2013). This can trigger accelerated ice flow of previously-buttressed outlet glaciers, as observed following the breakup of the Larsen B ice shelf in 2002 (Rignot et al.,

2004; Rott et al., 2011; Leeson et al., 2020). Third, ponding of surface meltwater overlying grounded ice can create ice-bed hydraulic connections via hydrofracture (Krawczynski et al., 2009), providing a mechanism by which surface-derived water can alter the basal hydrological system and affect the flow of grounded ice (Iken, 1981; Iken & Binschadler, 1986). This process has been inferred to occur on the Antarctic Peninsula (Tuckett et al., 2019), and could induce a fundamental change in Antarctic ice dynamics if it becomes widespread around Antarctica (Bell et al., 2018). Given the stated impacts that surface

water can have on ice sheet mass balance, it is important to understand how Antarctic surface hydrological systems operate and evolve through time (Arthur et al., 2020b). Antarctic-wide melt rates are projected to double by 2050 (Trusel et al., 2015), meaning that the influence of surface meltwater across Antarctica will become increasingly important for the mass-balance of the ice sheet as a whole (Bell et al., 2018).





Several methods have been developed to map supraglacial lakes (SGLs) from optical and Synthetic Aperture Radar (SAR)
satellite imagery. Methods include: i) optical image band reflectance thresholds (Stokes et al., 2019; Moussavi et al., 2020);
ii) supervised image classification techniques (e.g. Halberstadt et al., 2020); and iii) training machine learning algorithms
(Dirscherl et al., 2020). Though successful at identifying lakes, the application of these techniques has been limited in scope
due to a combination of time-expensive workflows, restricted data storage and computational resource limits. Automated
methods, combined with the advent of cloud-based computational platforms such as Google Earth Engine (GEE), provide the
opportunity to overcome these challenges, enabling large-scale and high-temporal resolution mapping of Antarctic surface
meltwater. The capabilities of GEE to map surface meltwater have been demonstrated in both Greenland (Lea & Brough,
2019) and Antarctica (Dell et al., 2020; Halberstadt et al., 2020), but GEE has yet to be used to generate pan-Antarctic results.
The majority of Antarctic SGL studies have mapped lakes from optical satellite imagery collected by passive satellite sensors
(Arthur et al., 2020b) due to its relatively high spatial resolution, the large archive of freely available imagery, and because
appropriate water detection techniques are well established and simple to implement (e.g. Moussavi et al., 2020). Optical
imagery is, however, detrimentally affected by spatially and temporally variable cloud cover, such that the resulting time series
of surface meltwater coverage are typically incomplete and inconsistent. Although investigation of controls on temporal and
spatial patterns in surface meltwater coverage requires analysis-ready data and is crucial to understanding the mass balance of
the Antarctic ice sheet, such data do not yet exist.

Here, we implement an image band reflectance threshold-based method (Moussavi et al., 2020) for SGL identification in GEE,
creating a fully automated method for mapping surface meltwater across Antarctica from Landsat imagery. We use both
Landsat 7 and Landsat 8 imagery, enabling us to create a multi-year time series of lake number and area from 2005-2020. We
apply a 'time window' approach, in which we present mapped results at bi-monthly temporal resolution over the duration of
each melt season. We also incorporate a novel approach to quantifying SGL coverage that accounts for variability in both
optical image coverage (e.g. region of interest coverage and Landsat 7 scan line corrector failure) and cloud cover. We
demonstrate our method across the Amery Ice Shelf region of East Antarctica, highlighting how the method will ultimately be
used to map meltwater at a pan-Antarctic scale. We present results showing the multi-year and seasonal evolution of surface
meltwater in the study region, and compare our results with climate data to investigate controls on surface melt extent.

## 2 Study Region

The Amery Ice Shelf (AIS) lies within an embayment of East Antarctica between the Prince Charles Mountains and Princess
Elizabeth Land. Covering an area of over 60,000 km$^2$, it is the largest ice shelf in East Antarctica and drains approximately 16
% of the East Antarctic Ice Sheet (Fricker et al., 2002; Spergel et al., 2021). The study area covers 188,828 km$^2$, of which 32
% is floating ice shelf, 68 % is grounded ice, and <1 % is exposed bedrock. The area has been divided into twenty-one 100 by
km tiles for processing in GEE (Fig. 1), and has been clipped to the coastline (Depoorter et al., 2013). The Amery Ice
Shelf region was selected for this study for the following reasons:



1) The AIS develops a large surface hydrological network of SGLs and surface streams on an almost annual basis (Spergel et al., 2021). Surface meltwater ponding is known to have occurred in this region for several decades (Phillips et al., 1998), hence we can be confident of generating a time series with significant amounts of surface water;

2) The AIS was one of the study areas used by Moussavi et al. (2020) to develop the meltwater mapping technique that is applied within this study. We can therefore be confident that the optical-band thresholds used by Moussavi et al. (2020) are appropriate for identifying surface water and masking out other land-surface types such as exposed bedrock and blue ice;

3) The region is a glaciologically important area of East Antarctica, due to the size of the ice shelf and the large catchment that it drains (Budd et al., 1966). Since surface melt can have a large impact on ice dynamic processes, it is important to understand how surface meltwater evolves in the region, and to determine long-term trends in surface water coverage. Although the AIS is currently largely resilient to hydrofracture (Lai et al., 2020), lake drainage events on grounded ice could influence ice flow dynamics in the near future (Tuckett et al., 2019).

4) The study area is large enough to be able to examine whether using our mapping approach at a pan-Antarctic scale is feasible. Processing capabilities within GEE are scaled to the number of lake polygons that are detected, meaning it takes longer to map areas with high numbers of SGLs. The AIS has a higher spatial density of SGLs that most regions in Antarctica (Stokes et al., 2019). By demonstrating that the method can successfully map SGL evolution over this region, we can be confident that it can be applied at a continental scale.





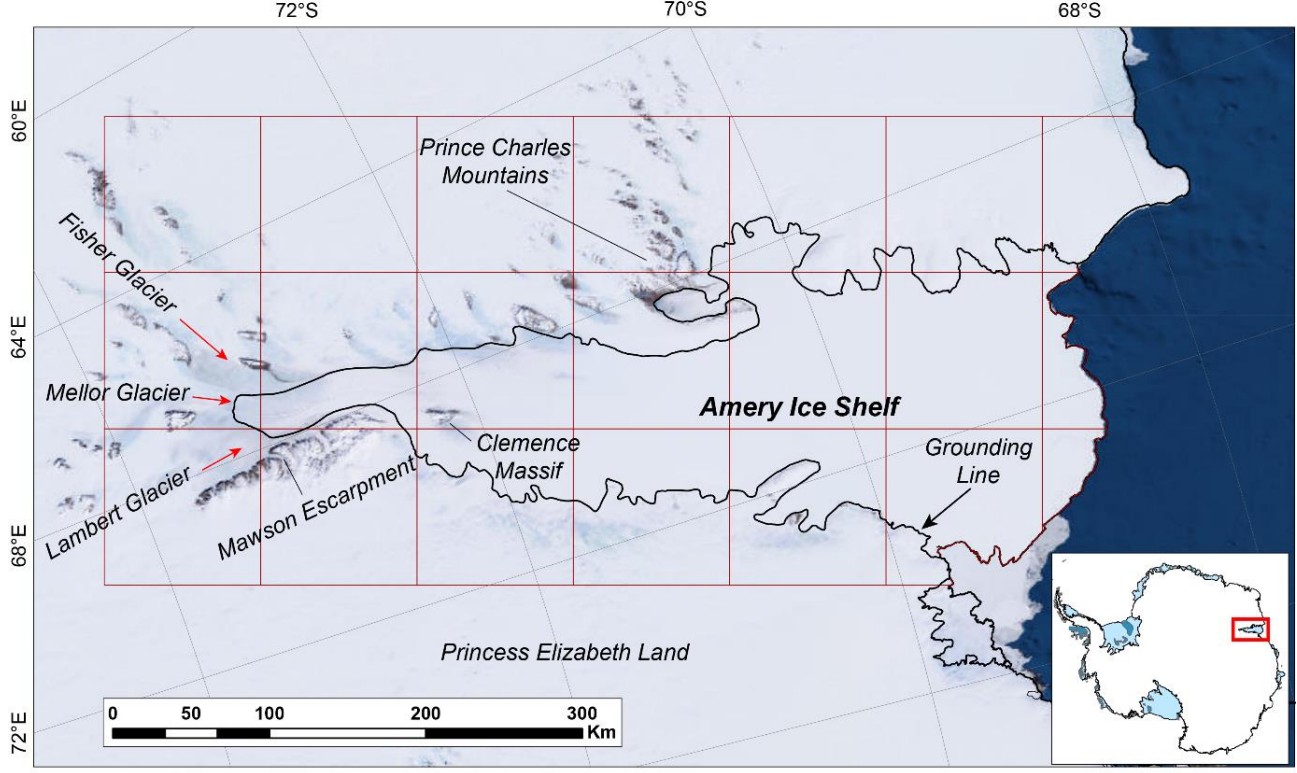

**Figure 1.** Study region over the Amery Ice Shelf, including an inset showing its location within Antarctica. The background

image is a 15 m TerraColour Antarctic Basemap (ESRI). The red boxes indicate the area over which melt was mapped, with

tiles representing twenty-one separate 100 by 100 km regions of interest (ROIs) for mapping within GEE. The black line

marks the grounding line based on the Depoorter et al. (2013) dataset. Red arrows indicate the flow direction of labelled

outlet glaciers.

## 3 Methods

Our method comprises four stages: (i) image data collection and filtering; (ii) identification of areas of surface meltwater; (iii)

image visibility assessment to quantify the area of surface water missed due to cloud cover and image data coverage; and (iv)

post-processing to generate polygon shapefile outputs and assign metadata. Stages 1-3 are undertaken within a single script in

GEE, whilst stage four is performed in Matlab (both codes available in the Supplement). Three inputs are required to run the

automated mapping tool in GEE: 1) A start and end date to define a date range for the image search; 2) A shapefile to specify

the total area over which lakes will be mapped; 3) The temporal resolution at which results will be generated, either as a

specified number of days or as a given number of time windows per month. Shapefile inputs are split into several ROI tiles to





limit the area that is mapped at once (Fig. 1), thus avoiding memory limit errors in GEE. The mapping procedure loops over all the ROI tiles (twenty-one tiles for the AIS region) within GEE to generate results across the study region. Below, we

describe the method over a single ROI tile.

## 3.1 Image Data Collection

Every Landsat 7 and 8 image covering any portion of our study region between 2005 and 2020 was used during analysis, totalling 4164 optical image tiles. In practice this resulted in Landsat 8 images being exclusively used beyond March 2013,

with Landsat 7 images used prior to this date. Images were not filtered by cloud cover to maximise the chances of detecting surface water. We used Level-1 Tier 2 Top of Atmosphere (TOA) Landsat image tiles which are directly available for analysis through the GEE data catalogue (https://developers.google.com/earth-engine/datasets/catalog/landsat, last access: 31 March 2021). TOA reflectance values are typically used for ice sheet studies in preference to raw digital numbers to ensure that pixel values are not influenced by differences in image acquisition conditions (Pope et al., 2016; Moussavi et al., 2020). Processing

was performed on a yearly basis, involving 16 runs of the GEE script (i.e. 2005-2020). For each GEE run, an image collection was generated from images that fitted the criteria of the specified time period and overlapped with the ROI. Images were additionally filtered to remove those with a sun elevation angle of less than 20°. Images with a sun elevation lower than this threshold value result in misclassification errors when using a band-threshold based approach, since in low light conditions surface water is not sufficiently spectrally different to be separated from features such as cloud and rock shadow (Halberstadt

et al., 2020; Moussavi et al., 2020).

## 3.2 Delineation of surface meltwater

We applied a surface meltwater detection method developed by Moussavi et al., (2020), who established threshold values to automatically identify surface water, cloud and rocks from Landsat 8 image bands (Table 1). The thresholds used in Moussavi et al. (2020) showed an accuracy of >95 % when identifying lake areas from Landsat 8 imagery, and results showed high levels

of agreement when compared with lake area data generated from other methods (Halberstadt et al., 2020). Whilst the thresholds developed by Moussavi et al. (2020) were designed specifically for Landsat 8, we found that the thresholds are highly successful when applied to Landsat 7 imagery, despite minor differences in the band wavelengths of the two satellites. Our analysis shows that there is an average agreement of ~90 % between Landsat 7 and Landsat 8 in the identification of surface water (see Figs. S1-3 in the Supplement for a comparison between Landsat 7 and Landsat 8).

As per the method of Moussavi et al. (2020), areas of exposed bedrock and seawater were removed from image tiles using a mask based on the thermal infrared (TIR) and blue bands. Cloudy pixels were removed using a combination of the Short-wave Infrared (SWIR) band, and the Normalized Difference Snow Index (Green-SWIR/Green+SWIR). Following application of these masks (Fig. 2), we then used the Normalised Difference Water Index (NDWI) to delineate areas of surface water. This is the most widely used technique for identifying water from optical imagery (Williamson et al., 2018; Arthur et al., 2020b),





and has been successfully used to map SGLs on both the Greenland (Pope et al., 2016; Moussavi et al., 2016; Williamson et
al., 2018) and Antarctic Ice Sheets (Stokes et al., 2019; Moussavi et al., 2020). See Table 1 for the threshold values used, and
Moussavi et al. (2020) for further details of the method. Once lake pixels were detected in each individual image tile, images
were assigned to time windows (Fig. 2). Lake masks from individual images within each time window were then combined to
create a single maximal lake mask for each time window.


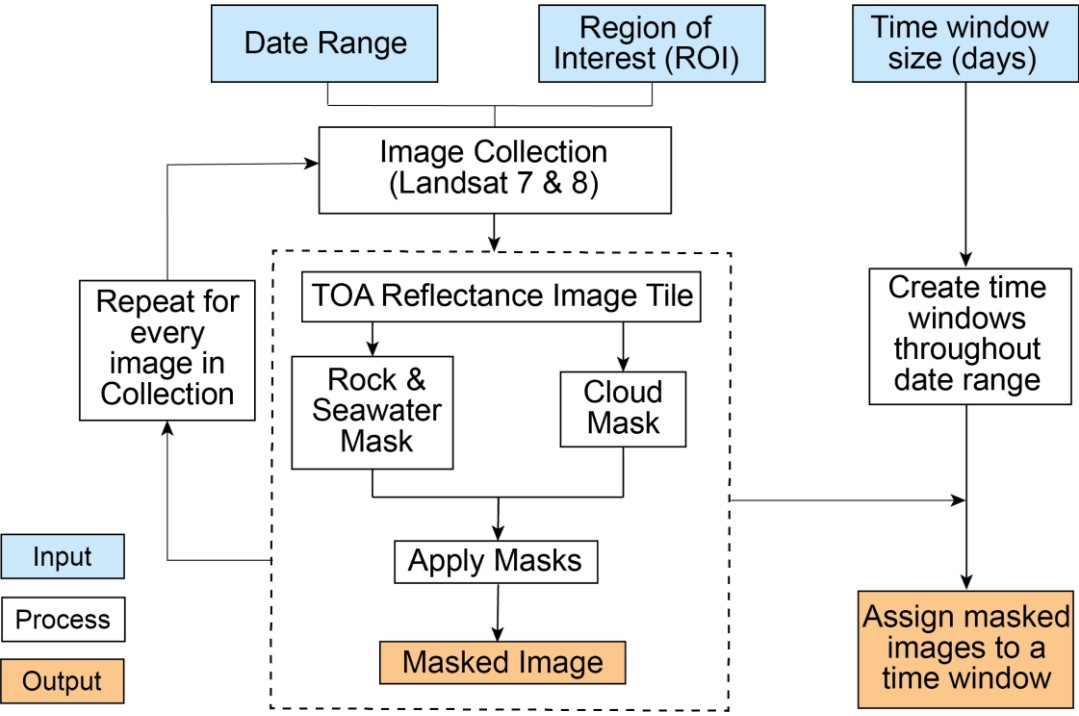

**Figure 2.** Flowchart illustrating the optical image masking steps taken within GEE, including the method by which images are
assigned to time windows. See Table 1 for the threshold values (Moussavi et al., 2020) used during each masking stage.


**Table 1.** Landsat image band reflectance thresholds (as detailed by Moussavi et al., 2020) applied during the masking and
surface meltwater detection stages within GEE.




| Classification | Thresholds applied |
|---|---|
| Rock/Seawater Mask | (TIRS1/Blue) > 0.35<br>Blue < 0.35 |
| Cloud Mask | (Green – SWIR1/Green + SWIR1) < 0.8<br>SWIR1 > 0.1 |
| Surface Meltwater | (Blue – Red/Blue + Red) > 0.1<br>(Green – Red) > 0.07<br>(Blue – Green) > 0.07 |

### 3.3 Lake Visibility Assessments

For images affected by cloud cover, mapped lakes from optical satellite data represent minimum estimates of true lake area. Though simple metrics of cloud cover per image are informative, they do not account for variability in meltwater extent and visibility within a time window. To account for the uncertainty in lake area due to these visibility issues, we developed a novel technique which estimates the potential maximum lake area likely if clouds were not present. To evaluate meltwater visibility over the duration of each time window, we therefore needed to assess two key aspects: i) A spatial assessment of the amount of ice visible within the intersection of each optical satellite image and each ROI, achieved by calculating an 'Image Visibility Score' (IVS) for every optical image (Fig. 3); and ii) A temporal assessment of the differences in meltwater extent between images within each time window. This second stage was achieved by calculating a 'Lake Pixel Contribution Score' (LPCS) for images within each time window (Fig. 3), enabling quantification of which images within any given time window contributed the most lake pixels to the overall output. These two metrics were then combined to estimate a 'Lake Visibility Percentage' (LVP) for each time window and ROI (Fig. 3).





**Figure 3.** Flowchart detailing the method used to conduct lake visibility assessments within GEE for each time window. Images (a-d) provide visual examples of selected stages, and are referred to within the flowchart. The different lake colours in (d) indicate which optical image each lake pixel has originated from (e.g. Red = Image 1, Black = Image 2 etc.). If the same





pixel is covered by water in more than one image within a time window, the image pixel with the highest NDWI value is promoted to the mosaicked image. Six images were used in this example, indicated by (x6).

### 3.3.1 Image Visibility Scores

An IVS was generated for every image tile that intersected each ROI, to provide a combined measure of ROI coverage and image visibility from cloud cover (Fig. 4). Each IVS represents the percentage of ice cover within the ROI that was visible in the optical image. First, a 'clear-sky' ice mask covering the study region was created in GEE from cloud-free images using the

rock mask thresholds stated in Moussavi et al. (2020). This enabled quantification of the area of ice covered by cloud in each image tile and facilitated removal of non-ice covered areas from IVS calculations, since we were only interested in areas where lakes could form on the ice surface. To calculate the IVS of a given Landsat image, both the cloud- and rock-masked optical image tile and the clear-sky ice mask were clipped to the extent of the ROI. These raster layers were then used to create a binary mask for each image which identified pixels within the ROI that were both visible (not obscured by cloud) and located

over ice. The areas (in km$^2$) of the ROI covered by both this 'visible over ice' mask and the clear-sky ice mask were then calculated within GEE. Each IVS was subsequently calculated following Eq. (1):

$$\text{Image Visibility Score (IVS)} = \frac{\text{Area of 'visible over ice' mask within ROI}}{\text{Area of 'clear-sky' ice mask within ROI}} \times 100 \qquad (1)$$

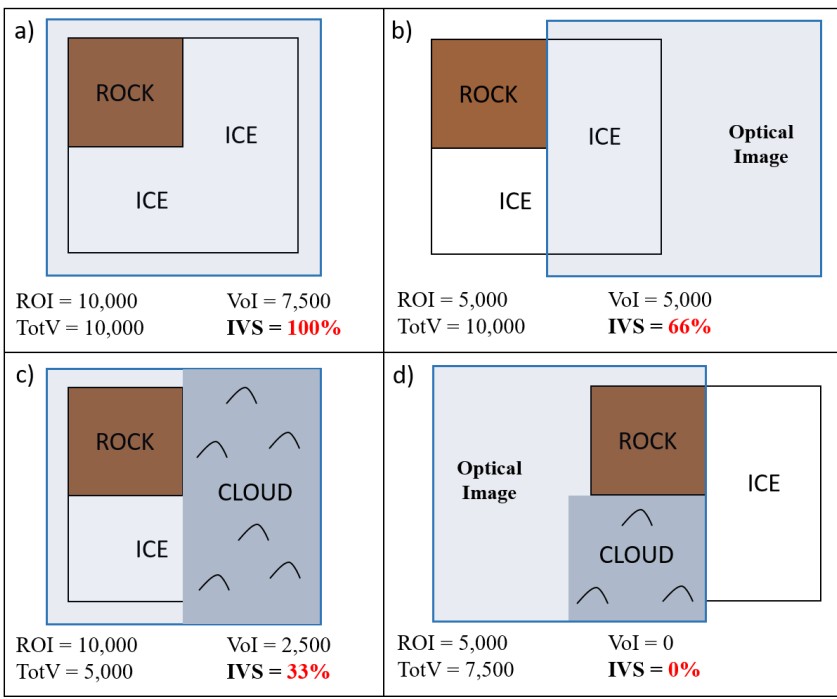





**Figure 4.** Schematic illustrations of four different image visibility scenarios, highlighting the IVS for each example. The black
square boxes show an ROI tile, representing a 100 x 100 km area. The same ROI tile is used in each example, comprising
7,500 km$^2$ of ice and 2,500 km$^2$ of rock. Blue boxes represent Landsat optical image tiles, which cover all (a & c) or half (b &
d) of the ROI. Optical images in (a) & (b) are cloud free, whilst images in (c) & (d) are partially cloud-covered. The numbers
below each example signify: i) ROI = the area (km$^2$) of the ROI that is covered by the optical image; ii) TotV = Total Visible.
The area (km$^2$) of the image that is visible (irrespective of the ROI); iii) VoI = Visible over Ice. The area (km$^2$) of ice within
the ROI that is visible; iv) IVS = Image Visibility Score. The IVS score in each example as a percentage. This is calculated by
dividing the 'visible over ice' (VoI) area by 7,500 (the area of the clear-sky ice in the ROI). Note how each IVS gives a
combined measure of ROI coverage, cloud extent and the proportion of ice within the ROI.

### 3.3.2 Lake Pixel Contribution Scores

Given that several images usually covered at least part of the ROI within a time window, it was important to know which of
them contributed the most to the detection of surface meltwater. To achieve a measure of this, we calculated a 'Lake Pixel
Contribution Score' (LPCS) for every optical image within each time window. Following the removal of cloud and rock areas,
we calculated the NDWI of images using the blue and red optical bands. A composite NDWI image for each time window was
then created whereby the highest NDWI value for each pixel was promoted (using the qualityMosaic function in GEE).
Following this, we clipped the NDWI composite to the ROI and applied the three thresholds (Table 1) recommended by
Moussavi et al. (2020) to identify surface meltwater pixels. Each image within a time window was assigned a unique ID prior
to mosaicking to identify from which image each lake pixel had originated. We achieved this by performing a frequency count
(ee.Reducer.frequencyHistogram) to determine the number of lake pixels within the ROI that were contributed by each
individual image. LPCSs were then calculated based on the proportion of lake pixels from each image that were used in the
composite lake mask for each time window. For example, an image LPCS of 0.4 meant that 40 % of the lake pixels identified
in the time window composite were extracted from that image.

### 3.3.3 Lake Visibility Percentages

For every image that contributed lake pixels within a given time window, the LPCS was multiplied by the IVS. These combined
scores were then summed to create a 'Lake Visibility Percentage' (LVP) for that time window (Table 2). This final measure
provided a representation of what area of meltwater coverage was likely to have been missed by our mapping approach. An
LVP of 100 % indicated that no lakes were missed (i.e. all of the ice surface was visible within the time window), whilst an
LVP of 50 % suggested that mapped results only accounted for half the likely true area of lakes. By performing this assessment
of lake coverage, we were then able to scale mapped lake area results up to 100 %, to attach an upper uncertainty bound to
minimum mapped lake areas. This approach assumes that every image pixel is equally likely to be covered by surface
meltwater, meaning scaled up results are only estimated values of lake area. In locations where SGLs are highly clustered, this





could result in over- or under-estimates. However, by performing the method over large ROI tiles and at a bi-monthly temporal resolution (meaning several images overlap each ROI per time window), this uncertainty is minimised.

**Table 2.** Example data highlighting how pixel contribution scores and their corresponding visibility scores are combined to create an overall 'Lake Visibility Percentage' for each time window.

| Image Number | LPCS | IVS ( %) | Combined Score |
|---|---|---|---|
| 1 | 0.11 | 66 | 0.11 x 66 = 7.26 |
| 2 | 0.40 | 81 | 0.40 x 81 = 32.4 |
| 3 | 0.23 | 45 | 0.23 x 45 = 10.35 |
| 4 | 0.08 | 99 | 0.08 x 99 = 7.92 |
| 5 | 0.02 | 12 | 0.02 x 12 = 0.24 |
| 6 | 0.16 | 100 | 0.16 x 100 = 16 |
| Lake Visibility Percentage | | | 74.17 % |

### 3.4 Post-processing steps

Mapped lake polygons and visibility statistics were exported as geoJSON files from GEE. Several post-processing stages were then undertaken in Matlab to convert the data into shapefiles, merge lake polygons between ROIs, and attach metadata. Shapefiles were firstly created (using the Antarctic Polar Stereographic projection) for every ROI tile and time window. ROI-specific shapefiles were then merged across the entire study region, to create one single dataset per time window. As part of this step, lakes split over ROI boundaries were joined together (Union), and inner polygons were 'cut' from outer lake boundaries in instances where an 'island' was present within a lake. We then calculated the area and geometric centroid of each cleaned polygon and applied an area threshold of two pixels, giving minimum lake areas of 1800 m$^2$ based on a Landsat resolution of 30 m. This filtered out noise from the raw output, likely associated with crevasse shadows or slush, whilst retaining enough data to include small lakes, especially those at high elevations that would have been missed with a higher area threshold value. Unlike some other studies (e.g. Stokes et al., 2019), we decided not to aggregate lake polygons in close proximity to each other, as tests showed this sometimes resulted in the false identification of large lakes in areas of meltwater filled crevasses. Finally, we attached selected metadata to each identified lake based on the geometric centroid of lake polygons. The Depoorter et al., (2013) grounding line dataset was used to label lakes as either 'grounded' or 'floating', whilst the elevation and surface slope of lake centroids were extracted from the Reference Elevation Model of Antarctica (REMA) database (100 m resolution) (Howat et al., 2019).



### 3.5 Comparison with Climate data

To provide an initial test of the extent to which climatic modelling can simulate surface meltwater ponding, we compared our lake area results with modelled snowmelt outputs from the Regional Atmospheric Climate Model version 2.3p2 (RACMO2.3p2) (van Wessem et al., 2018). RACMO2.3p2 has a horizontal resolution of 27 km and is coupled to an internal snow model which calculates surface meltwater production, refreezing, percolation, retention and runoff into the ocean. The model is forced by ERA-Interim reanalysis data (van Wessem et al., 2018). Mean monthly summed melt values were extracted

from the modelled data over a box covering the study region. RACMO2.3p2 snowmelt outputs serve as a boundary condition for meltwater availability, as the model does not specifically account for surface meltwater ponding. Moreover, it should be noted that RACMO2.3p2 locally resolves meltwater production based on 1-D model grid boxes, and hence does not account for the process of meltwater flowing from higher elevations onto the ice shelf (Spergel et al., 2021). Our analysis therefore offers a preliminary comparison between the two datasets rather than a full evaluation, which would require quantification of

lateral meltwater transfer and biases highlighted in van Wessem et al. (2018).

To explore the potential role of the large-scale atmospheric circulation on surface meltwater ponding in the study region, we investigated the influence of the Southern Annular Mode (SAM). The SAM is the main mode of extratropical climate variability across the Southern Hemisphere, and represents changes in the strength and position of the Southern Hemisphere westerly winds and storm tracks (Marshall & Thompson, 2016). We chose to compare our lake area results with the SAM

because of its known influence on Antarctic temperatures (Marshall & Thompson, 2016; Fogt & Marshall, 2020). We compared our results with austral summer values of the SAM index of Marshall (2003), obtained from (http://www.nerc-bas.ac.uk/public/icd/gjma/newsam.1957.2007.seas.txt. Last accessed: 31st March 2021).

## 4 Results

### 4.1 Evaluation of method

As shown by Moussavi et al., 2020, we find that the application of a band-thresholding technique within GEE is highly successful at rapidly identifying surface meltwater features over large areas and time periods. The method performed well over the whole study area at masking out areas of rock and cloud, whilst successfully identifying surface meltwater (Fig. 5). Very few false positive results were discovered, and the method performed well when differentiating lakes from areas of blue ice and shadow (Fig. 5). False negative results were rare, and mainly occurred where surface water was much darker in colour,

either due to sediment suspended within the water column or where lakes were very deep (over 5 m in depth). Instances of sediment laden water were confined to the immediate vicinity of rock outcrops, whilst lake depths very rarely exceed 4 m in the study region (Spergel et al., 2021). These misclassification errors thus had a minimal influence on results. As highlighted in Fig. 5, we found minimal difference in the performance of the method between Landsat 7 and 8 imagery (Fig. S1 in the Supplement).


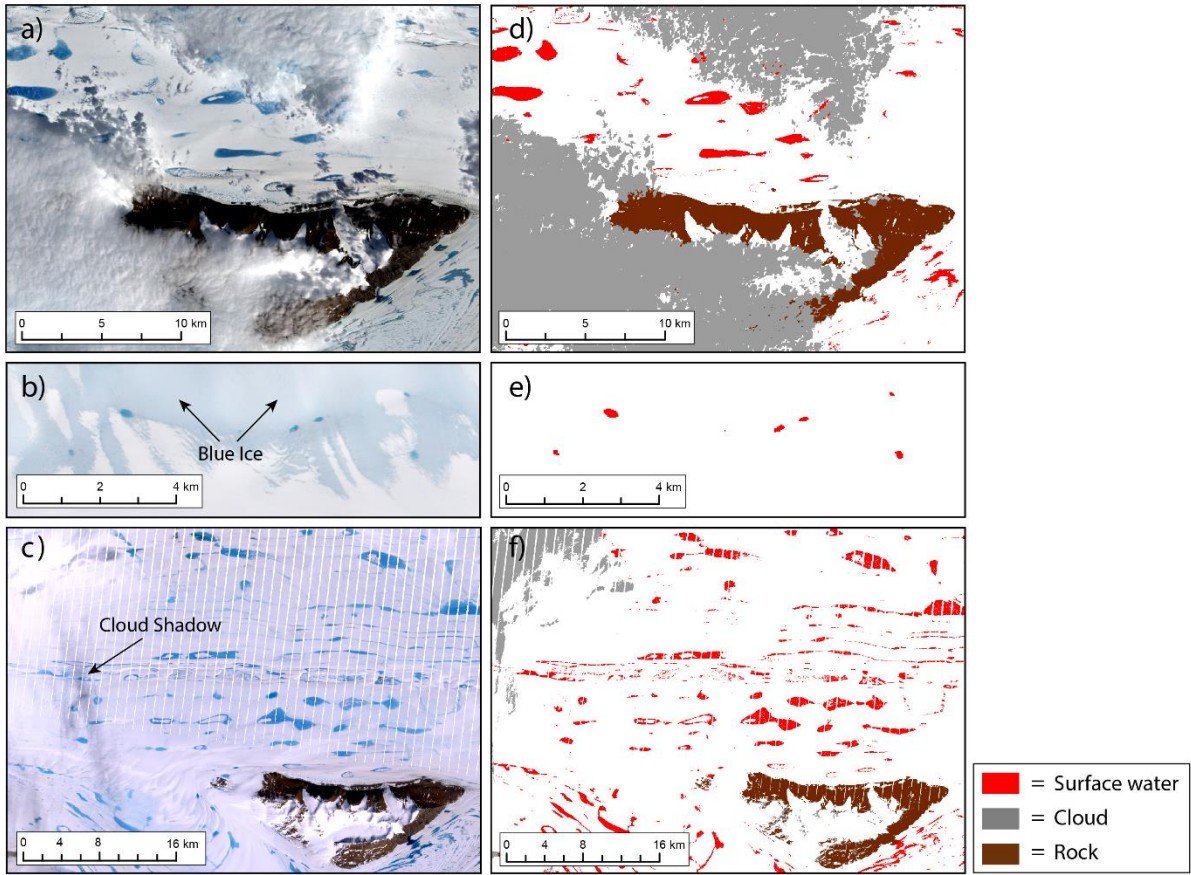

**Figure 5.** (a) Landsat 8 image from 25th January 2017 of the Clemence Massif; (b) Landsat 8 image from 1st January 2019, highlighting blue ice ~100 km south of Fisher Massif; (c) Landsat 7 image from 2nd January 2005, showing widespread surface lakes to the west of the Clemence Massif. Note the white stripes resulting from the failure of the Landsat 7 scan line corrector. (d-f) Automatic masking of cloud, rock and surface water from Landsat imagery.

LVPs ranged from 0-99.9 %, with a mean LVP of 50.4 % and a median LVP of 52.7 % across the whole dataset. However, there were large differences between LVPs from Landsat 7 and Landsat 8 images, largely due to data gaps present within Landsat 7 images as a result of the failure of the Scan Line Corrector (SLC). The median LVP from time windows using Landsat 7 imagery was 43.5 %, compared to 61.6 % when Landsat 8 images were used. By using LVPs to generate maximum lake area estimates, we were able to account for lake area underestimations resulting from data gaps in Landsat 7 imagery. On average, incorporating LVPs into lake area estimates resulted in a 58 % increase in lake area per ROI and time window when using Landsat 7, and a 42 % increase when using Landsat 8 images. When results were aggregated to generate cumulative lake area estimates per melt season, maximum potential lake area estimates were 42 % greater than mapped values on average across the entire study period.



## 4.2 Spatial distribution of SGLs

We find that SGLs form on inland areas of the AIS where the ice shelf is narrowest, and on portions of grounded ice within close proximity to the grounding zone (Fig. 6). On average, ~70 % of total lake area within the study region exists on the ice shelf and ~30 % on grounded ice. In high melt years, SGLs are widespread across the width of the ice shelf between ~72-73° S, and along the Prince Charles Mountains side of the ice shelf to around 71° S. Very few lakes form on the ice shelf interior further north than this latitude, although a cluster of lakes sometimes form in a sub-inlet of the ice shelf near the Prince Charles Mountains (Fig. 6). Lakes on the ice shelf most frequently form on the south-east side of the Clemence Massif, and on the eastern side of the Fisher Massif (Fig. 6). SGLs in these locations are typically elongate in shape, and are connected by surface streams and channels to form a distributed surface drainage network. During high melt years, the largest lakes are found along the central flowline of the ice shelf below 71° S; the largest mapped lake in our study had an area of 107 km$^2$ in January 2005. However, these central lakes vary greatly in size and occurrence between melt seasons, whilst lakes nearer the grounding zone and next to areas of exposed bedrock form more frequently (Fig. 6).

SGLs on grounded ice predominantly form within approximately 20 km of the grounding zone, and are particularly abundant along a 200 km stretch of the Princess Elizabeth Land ice shelf boundary between 70 – 72 ° S (Fig. 6b). Lakes in this region, which can be up to 6 km$^2$ in area, typically form in the same location on an annual basis. Whilst the spatial extent of lakes varies between years, we noted several lakes in this region that formed in the same location during all 14 of the complete melt seasons studied (Fig. 6b). No large lakes form on the three main glaciers, which feed the southern-most portion of the ice shelf, but extensive areas of meltwater-filled crevasses are often observed on Lambert glacier.



**Figure 6.** Spatial distribution of SGLs over the study region, showing the recurrence frequency of surface meltwater between 2005 and 2020. The maximum recurrence frequency is 14 years, due to the exclusion of the 2004/05 and 2018/19 melt seasons. Pixels were assigned values of 1 (melt) or 0 (no melt) per year, based on the occurrence of surface water at any stage during each melt season. Pixels were then summed to derive recurrence frequency. The linear light blue feature near the ice shelf calving front is a misclassification error associated with a large calving event that occurred in September 2019 (Walker et al., 2021).



Surface meltwater is found up to elevations of ~1500 m, with the highest confirmed lake (with a minimum area threshold of 1800 m$^2$) existing at 1591 metres above sea level (m a.s.l.). Lakes are most common at low elevations, with the greatest lake area totals identified between 100-200 m a.s.l. This is the elevation band that covers the majority of the southern part of the

345 ice shelf. The majority of the northern half of the ice shelf lies below 100 m a.s.l. but there is low runoff and ponding in this region (Fig. 6). Average lake size decreases with an increase in elevation, with the majority of surface meltwater above ~600 m a.s.l. existing in the form of small, isolated ponds within crevasses fields (mostly on Lambert Glacier). However, larger SGLs (up to ~5 km$^2$ in area) are common at elevations up to 500 m a.s.l. on sections of grounded ice in Princess Elizabeth Land. Lake areas are greatest between 100 and 200 m a.s.l. during all five months of the melt season (Fig. 7), regardless of

350 annual variations in absolute melt supply. We do, however, notice slight differences in the distribution of lake area across elevation bands between high and low melt years. During low melt years, total lake area is more evenly distributed across elevations ranging between 100 – 400 m a.s.l. (Fig. 7b), whereas in high melt years, lake surface areas are more concentrated between 100 – 200 m a.s.l. (Fig. 7a).

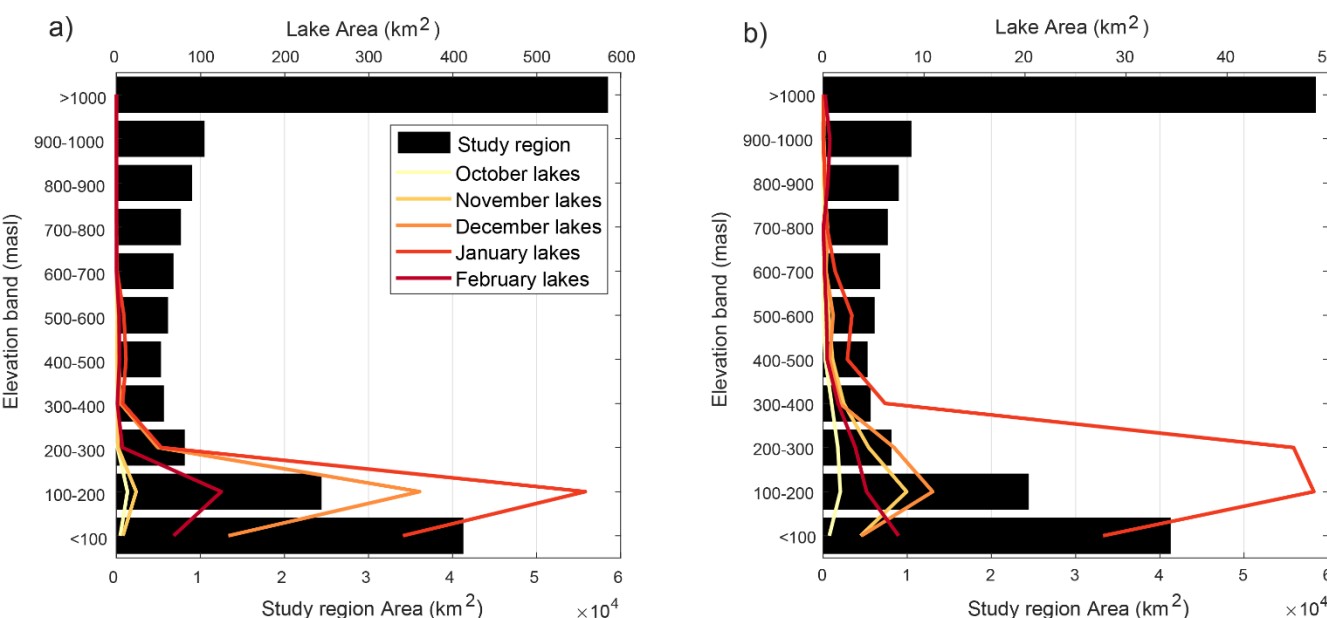

**Figure 7.** Averaged total lake areas per month by elevation bands, for a high melt season (a, 2005/06) and a low melt season (b, 2015/16). Black horizontal bars show the hypsometry of the study region. Note the total lake area is an order of magnitude greater during the high melt year (see lake area scales).



### 4.3 Temporal evolution of surface meltwater

The seasonal and multi-year evolution of lakes for the Amery region is shown in Fig. 8. There is high variability in both the number and total areas of lakes observed between melt seasons, but we do not observe an overall increasing or decreasing trend. The highest number of lakes were observed during the 2016/17 melt season, during which the cumulative total number of lakes exceeded 100,000. By contrast, less than 30,000 lakes were cumulatively observed during both the 2010/11 and 2011/12 melt seasons. There is a strong correlation ($r = 0.81$, $p = 2.1 \times 10^{-32}$) between the number of lakes and the total lake area for individual time windows. In addition to having the highest number of lakes, the 2016/17 season also had the highest cumulative lake area, with an estimated (based on lake visibility corrected scores) maximum lake area total of 5179 km$^2$. High lake area totals were recorded during the 2005/06 season, despite only having the sixth highest number of lakes.

The highest lake area totals during an individual time window were identified during the 2004/05 and 2005/06 melt seasons. During the first half of January 2005, surface meltwater covered an estimated maximum total area of 2814 km$^2$. This was almost three times greater than the average total lake area for the first half of January (963 km$^2$ for maximum estimates) throughout the study period. The seven year period between late 2006 and early 2013 was characterised by low levels of surface meltwater (Fig. 8). The average estimated cumulative lake area per season during this time period was 1062 km$^2$. This was around three times lower than the equivalent average of 2997 km$^2$ between 2014 and 2020 (excluding 2018/19 due to incomplete data availability), despite the 2015/16 melt season having very low areas of lake coverage. Multi-year trends in the number of lakes followed a similar pattern, although the drop in the number of lakes during 2015/16 was less significant than in equivalent lake area measurements.

Clear seasonal patterns of lake numbers and areas can be observed within each melt season (Fig. 8). Between October and early December, total lake areas were typically very low, with any meltwater forming in crevasses or pooling in small lakes close to exposed bedrock. For all studied years, there was a sharp increase in total lake area during the second half of December, including in melt seasons when absolute lake area values were relatively low. On average, total lake area increased by an order of magnitude during this time window compared to the first half of December. Lake area coverage typically continued to increase into the first half of January, when maximum lake areas for the melt season were most commonly observed. Peak lake area totals were experienced during the first half of January on eight out of the fourteen occasions for which data were generated throughout the entire melt season (Table 3). In low melt years, it was more common for lake areas to peak later in the melt season, usually during the second half of January and on one occasion (2009/10) during the first half of February. In most years, total lake area decreased through late January and early February, and by the second half of February, most lakes had frozen over. The average estimated total lake area for late February was 97 km$^2$, compared with 348 km$^2$ during the first half of the month. Despite these seasonal trends in total lake area, we did not observe a shift in meltwater cover to higher elevations throughout each melt season (Fig. 7).


**Figure 8.** Time series showing the temporal evolution of lakes over the Amery Ice Shelf region between 2005 and 2020. (a) Number of lakes per time window and cumulatively over each melt season; (b) Observed minimum and estimated maximum lake area per time window, in addition to seasonal cumulative totals; (c) Monthly modelled melt sum over the study region, from RACMO2.3p2. Note that cumulative totals are not included for 2004/05 and 2018/19 due to incomplete data availability over these melt seasons.

**Table 3.** Descriptive statistics for the time window with the greatest total lake area, for each melt season included in the study.





| Melt season | Time window of highest total lake area | Largest lake area (km$^2$) | Standard deviation of lake area | Elevation of 95th percentile lake (min 4 pixels) (m a.s.l.) | % Lake Area Grounded |
|---|---|---|---|---|---|
| 04/05 | 1-15 January 2005 | 107.1 | 1.08 | 430 | 18 |
| 05/06 | 1-15 January 2006 | 57.5 | 1.03 | 389 | 13 |
| 06/07 | 16-31 January 2007 | 4.8 | 0.11 | 469 | 53 |
| 07/08 | 1-15 January 2008 | 5.1 | 0.11 | 434 | 43 |
| 08/09 | 1-15 January 2009 | 7.1 | 0.13 | 422 | 52 |
| 09/10 | 1-14 February 2010 | 17.9 | 0.28 | 459 | 32 |
| 10/11 | 16-31 January 2011 | 2.9 | 0.09 | 368 | 53 |
| 11/12 | 16-31 January 2012 | 4.9 | 0.16 | 348 | 36 |
| 12/13 | 16-31 January 2013 | 3.0 | 0.10 | 332 | 28 |
| 13/14 | 1-15 January 2014 | 21.6 | 0.27 | 406 | 47 |
| 14/15 | 1-15 January 2015 | 52.2 | 0.59 | 382 | 22 |
| 15/16 | 1-15 January 2016 | 1.8 | 0.05 | 405 | 64 |
| 16/17 | 1-15 January 2017 | 32.0 | 0.40 | 436 | 28 |
| 17/18 | 1-15 January 2018 | 7.4 | 0.12 | 418 | 56 |
| 18/19 | 16-31 December 2018 | 15.4 | 0.20 | 451 | 39 |
| 19/20 | 16-31 January 2020 | 23.2 | 0.29 | 452 | 33 |


## 4.4 Comparison with climate data

We compared our lake area results with monthly surface snowmelt rates from RACMO2.3p2 to investigate the relationship between observed and modelled results. There is strong positive correlation between the seasonal totals of the two datasets (r = 0.76, p = 0.002), showing that the RACMO model captures the temporal variations in melting indicated by lake observations reasonably well (Fig. 9). The two melt seasons with the highest cumulative total lake area (2016/17 and 2005/06) also had the highest seasonal melt estimates. However, the seasonal melt total for 2005/06 was 23.7 kg m$^2$ greater than the 2016/17 estimate, despite displaying very similar cumulative lake areas. The biggest discrepancy between the two datasets was in 2014/15 when modelled melt rates were low, whereas the cumulative lake area was the third highest throughout the study period.

Figure 8c reveals minor inter-annual variations in both the spread and the maximum estimates of modelled melt rates. Monthly RACMO melt totals were highest during December in most of the study years, but peak melt was modelled to have occurred during January in six melt seasons. In years when maximum melt was modelled to have occurred during December, total lake area typically (75 % of the time) peaked during the first half of January, indicating a lag between peak melt and peak lake storage of ~15-30 days. Similar lag times were observed in years when modelled melt values were highest in January, with total lake area in these years most commonly peaking in either the second half of January or early February (Table 3). The duration of high (>20 kg m$^2$) melt rates also varied between years. In 2005/06, high melt rates were experienced over a single





month (December), whilst remaining very low during other months of the melt season. This matches well with the lake area data for that year, where a sharp increase in total lake area was observed between mid-December and mid-January, before rapidly dropping again by the end of January. In some years, maximum melt rates were sustained over both December and January, although absolute values of melt rate were usually lower in these years. In 2012/13, for example, the maximum

monthly melt estimate was 21.0 kg m$^2$, but because this level of relatively high melt was sustained over a period of two months, the seasonal melt total was the fourth highest during the study period (Fig. 9).

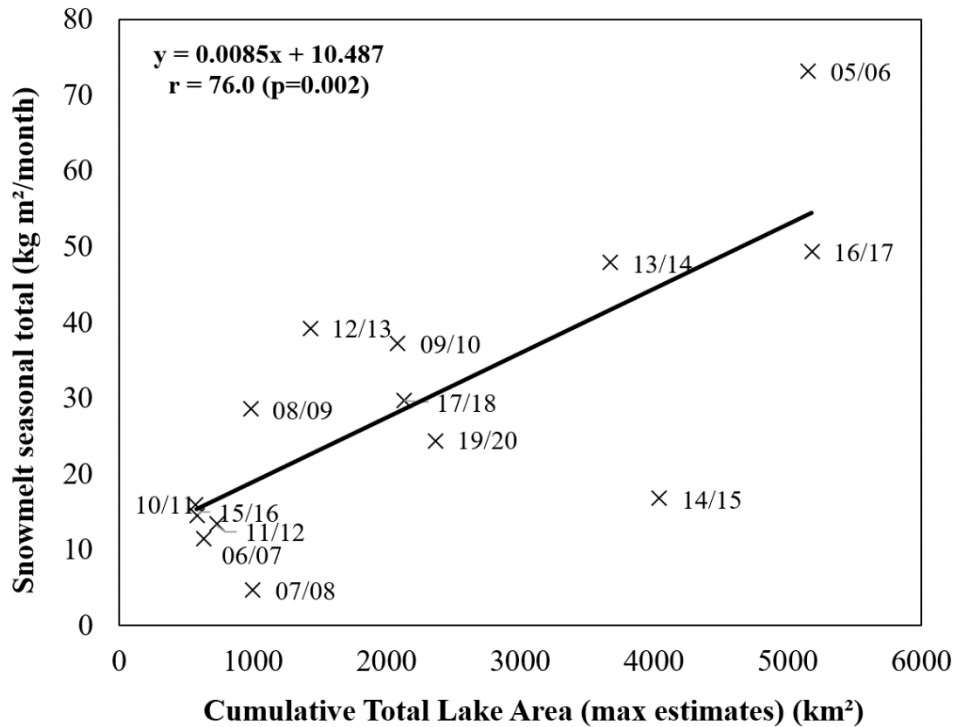

**Figure 9.** Scatter plot and correlation statistics of the relationship between cumulative RACMO melt and cumulative lake area

over the study region per melt season.

To investigate the extent to which large scale variability in Antarctic climate influences surface meltwater area, we correlated our lake area results against the SAM Index (Fig. 10). We find that there is a significant negative correlation (r = -0.54, p = 0.029) between total lake area and the SAM index for austral summer months. Melt seasons with a negative summer SAM

index correlated with years when total lake areas were greatest, whilst years with a positive summer SAM index were associated with low accumulations of surface meltwater. The SAM index was below minus one on two occasions throughout the study period (2005/06 and 2016/17), the same two years that we observed the greatest cumulative lake areas (excluding the 2004/05 melt season where data were only available during the second half of the melt season). Years with a positive SAM





index of two or more were characterised by low surface melt cover, with the notable exception of the 2014/15 season. This

melt season was associated with the highest SAM index of the whole study period, yet had the fourth highest cumulative lake

area total.

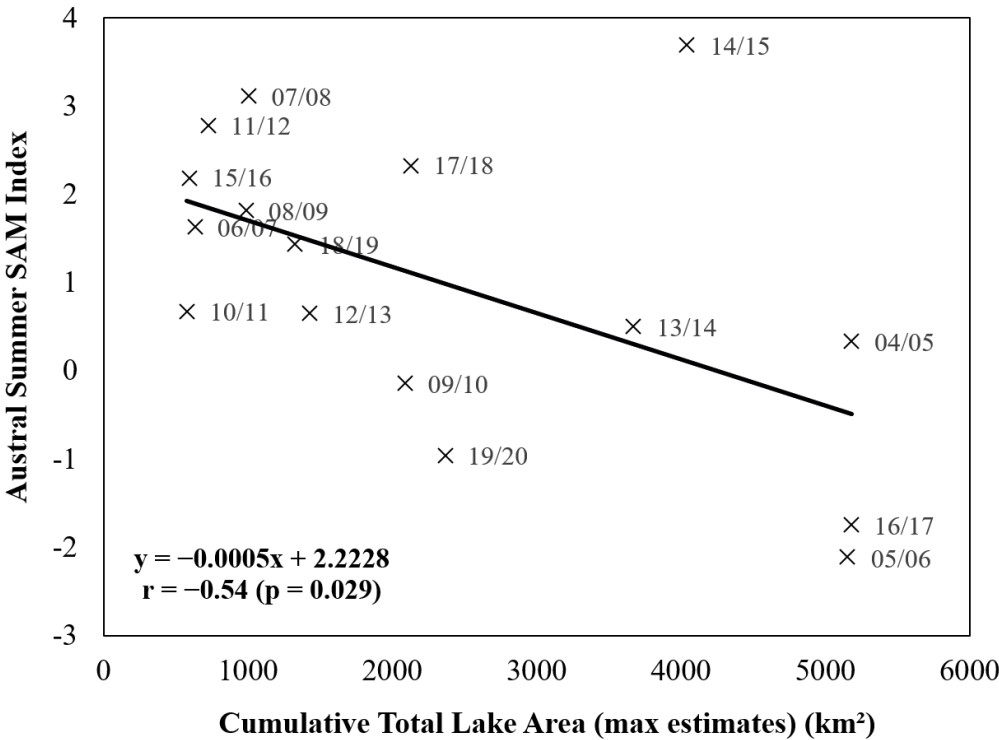

**Figure 10.** Scatter plot and correlation statistics of the relationship between the austral summer SAM index and cumulative total lake area per melt season.

**5 Discussion**

**5.1 Improvement in the assessment of surface melt extent**

In this paper, we have overcome two key factors which previously restricted the generation of robust high-resolution time series of SGL extent from optical satellite imagery. First, by incorporating a threshold-based method for lake detection within GEE, with results generated by time windows, we have created a fully automated method for generating lake area time series

that is quick and simple to run. The majority of SGL mapping studies in Antarctica have been limited in spatial and/or temporal resolution, partly due to methodological constraints relating to the computational expense of processing large imagery datasets. Despite having a relatively high spatial density of SGLs compared to most other areas of Antarctica (hence reducing the speed of processing within GEE), we were able to map an area of >185,000 km$^2$ over a 15-year time period in less than a week of





wall-clock time. This rapid processing opens up the possibility of future studies to investigate surface meltwater evolution over

vastly increased spatial and temporal scales, compared to what would be possible using manual or semi-automated methods. The method requires minimal inputs and user intervention (file transfers are required between the GEE and Matlab automated stages), meaning it can be quickly adapted to generate lake area time series for other regions of Antarctica, and ultimately a pan-ice sheet study. By using a time-window approach whereby the temporal resolution can be varied, the method could be used to investigate surface meltwater processes at range of temporal resolutions (depending on image availability).

Second, our SGL mapping procedure incorporates a robust new method for assessing image visibility, enabling us to account for variability in cloud cover and image data coverage when generating time series. Whilst multiple studies have provided Antarctic SGL area and volume estimates from optical mapping (Arthur et al., 2020; Dell et al., 2020; Moussavi et al., 2020), accounting for low image visibility from cloud cover has remained the primary limiting factor in creating a continuous and consistent time series (Moussavi et al., 2020). Furthermore, reported SGL areas and volumes based on optical mapping likely

underestimate ground-truth meltwater extent, since very few optical images are entirely cloud-free. Here, we performed image visibility assessments on every image analysed, enabling us to quantify levels of uncertainty for lake area results. Maximum lake area estimates, which incorporated visibility assessments, increased mapped lake areas for time windows on average by approximately 50 %. This highlights the importance of accounting for image visibility when reporting lake area results, especially when working with Landsat 7 imagery (due to the SLC failure) or mapping frequently cloud-covered regions, such

as the Antarctic Peninsula (van Wessem et al., 2016).

## 5.2 Spatial distribution of surface melt on Amery Ice Shelf

Surface lakes are often widespread on inland sections of the ice shelf during austral summer months, whilst almost no SGLs form on the northern half of the ice shelf nearer the ocean. The spatial distribution of surface lakes on the AIS is strongly influenced by variations in firn air content across the study area, as similarly observed across other ice shelves in Antarctica

(Lenaerts et al., 2017; Arthur et al., 2020a; Dell et al., 2020). The lack of surface meltwater ponding in the northern half of the study region (Fig. 6) is likely a consequence of high rates of snow accumulation near the calving front (Budd, 1966). A thick snowpack near the ice front has large pore spaces within the firn layer, meaning surface meltwater can percolate downwards and be accommodated within the pore spaces (Bell et al., 2017). By contrast, low accumulation rates further inland on the ice shelf likely result in a lower firn air content, meaning the firn layer becomes saturated with meltwater more quickly causing

ponding of surface water (Bell et al., 2017; Arthur et al., 2020a). Cycles of melting and re-freezing increase the grain-size of particles within the firn layer, reducing the albedo of the surface compared to fine-grained fresh snow (Zwally & Fiegles, 1994; Phillips, 1998). This can induce a positive feedback whereby previously melted areas are more likely to experience further melting, due to the increased absorption of short-wave radiation associated with low albedo surfaces (Kingslake et al., 2017). It is possible that this feedback is further be enhanced by the presence of ice slabs and lenses which can from beneath areas of

intermittent pond formation (Hubbard et al., 2016). These dense layers of ice inhibit meltwater percolation, and can be several



degrees warmer than ice that hasn't undergone lateral heat fluctuations that result from the melting and refreezing of ice (Hubbard et al., 2016).

The clustering of surface lakes around the grounding line at southern latitudes of the AIS can further be explained by the influence of katabatic winds. Near-surface air temperatures in coastal regions of East Antarctica are strongly influenced by
katabatic winds which originate from the ice sheet's interior (Lenaerts et al., 2017). These winds, which are commonly strong and directionally persistent (Lenaerts et al., 2017), generate localised surface and atmospheric conditions that are conducive to surface melting. Katabatic winds warm adiabatically as they flow down surface slopes, disrupting the natural temperature inversion and resulting in warmer, more humid air adjacent to the ice surface at the break in slope of the grounding zone (Doran et al., 1996). These atmospheric conditions, combined with the occurrence of low surface slopes on the ice shelf, optimise the
local environment for meltwater ponding, resulting in SGL formation around the grounding zone of Antarctic ice shelves (Arthur et al., 2020b). Particularly high numbers of lakes are observed on the narrowest part of the AIS, as this is likely the focal point for katabatic winds that are channelised, and hence strengthened, down Lambert, Fisher and Mellor glaciers (Zwally & Fiefles, 1994). Furthermore, increased numbers of flow stripes in this narrow section of the ice shelf provide greater surface roughness within which lakes can form (Ng et al., 2018). Our results show that lakes form to lower latitudes along the Prince
Charles Mountains side of the ice shelf compared to the Princess Elizabeth Land margin (Fig. 6). We suggest this is because katabatic winds continue to be channelised by the Mawson escarpment once on the ice shelf, causing them to naturally flow out along the western margin of the ice shelf. Once the ice shelf widens and is no longer as confined by topography, the winds likely weaken in strength, thus negating the localised warming effect and limiting lake growth.

Strong katabatic winds can also erode the surface snow layer within which melt could be stored, exposing highly compacted,
less permeable surfaces. Continued wind scouring around the grounding zone can expose areas of blue ice, which have a lower albedo (~0.57) than refrozen snow (~0.7) (Lanaerts et al., 2016). The presence of blue ice, in addition to the high number of low-albedo nunataks that surround the inland portion of the AIS, increases net surface absorption of solar energy, providing a localised warming effect and enhancing surface melt rates (Kingslake et al., 2017). Surface melt rates on other ice shelves in Antarctica, such as Roi Bedouin and Shackleton, have been shown to be strongly controlled by melt-albedo feedbacks
(Lenaerts et al., 2017; Jacobs et al., 2019; Arthur et al., 2020a; Dell et al., 2020). Our results support these findings, as we observe a clear spatial association between low albedo surfaces and areas of high lake occurrence, such as the large number of lakes that form annually next to the Prince Charles Mountains (Fig. 6). The spatial distribution of surface meltwater in the study region is hence closely controlled by melt-albedo coupling between exposed bedrock, blue ice and surface melting (Kingslake et al., 2017).

On both grounded and floating sections of the study region, lakes typically form in the same location on an annual basis (Fig. 6). Surface topography controls the hydrological routing of surface water, resulting in the ponding of water in small hollows and basins (Bell et al., 2017). Longitudinal surface structures on the ice shelf surface, caused by lateral compression and longitudinal extension of ice (Glasser et al., 2015; Ely et al., 2017), channelize surface meltwater downstream, likely explaining the elongate shape of lakes observed on the ice shelf. Variations in the downstream extent of lakes between years are therefore





likely to partly be a consequence of variable melt supply (Spergel et al., 2021). The distribution of surface basins on grounded
       ice is controlled by subglacial topography, meaning lakes can form annually in fixed surface depressions (Echelmeyer et al.,
       1991; Igneczi et al., 2018).

## 5.3 Temporal variation in ponded surface meltwater on the Amery Ice Shelf

There is a clear intra-seasonal pattern of total lake area; it remains low through the early part of the melt season, before rapidly
       increasing during late December and reaching a maximum in January (Fig. 8; Table 2), and then decreasing sharply during
       February. This matches with results from scatterometer studies which show large decreases in backscatter values over the AIS
       in January, indicating a rapid increase in the intensity of surface melting (Oza et al., 2011). The sudden increase in lake area
       (up to an order of magnitude increase within half a month) is likely a consequence of the hypsometry of the study region. Over
35 % of the study region (~65,000 km$^2$) lies at an elevation lower than 200 m a.s.l., meaning that a minor increase in
       temperature increases melt potential over a vast area of ice. This contrasts with the typical hypsometry of the Greenland Ice
       Sheet, where relatively steep slopes at the ice sheet margin mean that an equivalent rise in temperature would initiate melting
       over a much smaller area (McMillan et al., 2007; Sundal et al., 2009). The large lake area contribution from low elevations
       possibly explains why we do not observe a major elevation shift in peak area contribution throughout the melt season (Fig. 7),
as the signal from the ice shelf masks any changes in total lake area contribution at higher elevations. Following the initial
       appearance of meltwater ponds, overall lake area is likely further enhanced by positive feedbacks, whereby lowered surface
       albedo from melting promotes further melting. Furthermore, the development of surface streams enables lateral transfer of
       surface water, rapidly increasing the spread of water across the ice shelf surface (Kingslake et al., 2017). Sharp decreases in
       lake area during February are presumably indicative of the widespread freezing of SGLs, although evidence of lake drainage
events have also been observed in the region (Fricker et al., 2009; Pan et al., 2020; Spergel et al., 2021).
       There is a strong association between annual cumulative lake area and the summer SAM index (Fig. 10), suggesting that ice-
       shelf wide annual variations in lake area cover are influenced by large-scale climate variability. Phases of the SAM naturally
       oscillate on a multi-decadal timescale (Picard et al., 2007), possibly explaining the observed multi-year phases between periods
       of low and high lake area coverage (Fig. 8). When SAM is in a positive phase, air temperatures are typically higher over the
Antarctic Peninsula and lower over the rest of the continent, whilst the reverse is the case during a negative SAM phase
       (Marshall & Thompson, 2016; Turner et al., 2020). Our results broadly support this relationship, as observed by the statistically
       significant negative correlation between lake area and summer SAM index (Fig. 10). For example, the seven-year period
       between 2006 and 2013, which was largely characterised by positive summer SAM indexes, coincided with low annual
       cumulative surface meltwater coverage. The only year during this period with a negative summer SAM index (where we would
expect slightly warmer temperatures) was in 2009/10. This melt season had the highest cumulative lake area of this seven-year
       period, suggesting that the summer SAM index has a direct effect on melt rates on an annual basis. This wider climatic control
       on SGL formation suggests that the AIS has an abundance of basins within which meltwater can be accommodated, resulting



in a linear relationship between melt rates and SGLs (Fig. 9). This may not necessarily be the case in other regions of Antarctica, where steeper topography may limit the number and size of depressions able to host meltwater, thus resulting in
enhanced surface runoff and a non-linear relationship between melt and SGL area.

There was high variability in the austral summer SAM index from 2013-2020, ranging from -1.75 in 2016/17 to 3.69 in 2014/15. In general, lake areas followed the broad pattern we would expect based on their association with the SAM throughout this time period, with the main exception of the 2014/15 melt season. Large lakes formed during this melt season, despite there being a negative SAM and low melt rates predicted by RACMO. Greater than expected meltwater ponding in this year can be
explained by enhanced scouring of the ice shelf surface by strong katabatic winds. Following a snowfall event in late November 2014, large areas of low-albedo blue ice were exposed on the ice shelf by mid-December (Fig. 11a,b), suggesting strong wind-scouring throughout the first half of December. Between the 21$^{st}$ – 28$^{th}$ December, the ice shelf was transformed from being almost entirely lake-free to widely covered by SGLs (Fig. 11c). The following melt season, by contrast, snow cover persisted across most of the ice shelf throughout December (Fig. 11e), meaning any meltwater could be accommodated within the firn
pack rather than ponding as surface water.

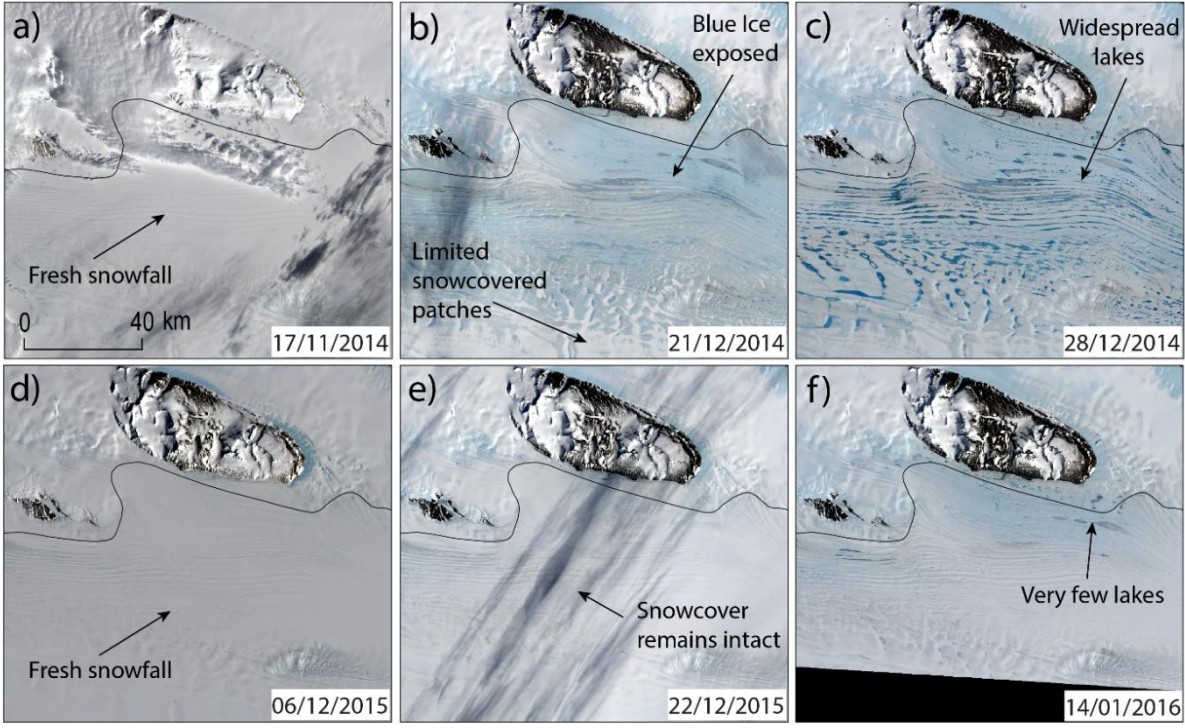

**Figure 11.** Landsat 8 images showing the evolution of the ice shelf surface to the east of the Fisher Massif in the 2014/15 (a-c) and 2015/16 (d-f) melt seasons.



The formation and extent of SGLs are highly sensitive to minor fluctuations in surface air temperature (Langley et al., 2016).
During December 2014, the ice shelf was pre-conditioned as a low-albedo, impermeable surface, optimising the conditions
required for surface meltwater ponding. Given this, it is likely that a transient increase in air temperature, possibly induced by
a strong katabatic event, could have resulted in a large change in surface melt characteristics. Surface melt rates depend on all
terms of the surface energy balance (Oza et al., 2011), meaning air temperature is not the sole factor in determining surface

melt rates. Whilst RACMO modelled melt estimates include a surface albedo parameterization, melt-albedo feedbacks are
difficult to resolve as the model does not specifically include a representation of blue ice. Previous studies have shown that
RACMO often under predicts meltwater production in areas of Antarctica where blue ice is warmed by katabatic winds (Trusel
et al., 2013; Leanaerts et al., 2017). This possibly explains why there were such major differences in lake area coverage between
2014/15 and 2015/16, despite RACMO seasonal snowmelt estimates differing by only ~2 kg m$^2$ (Figs. 8 & 9). Jakobs et al.

(2019) found that surface albedo was the main difference in ice surface characteristics between high and low melt years on the
Ekstrom ice shelf, supporting our hypothesis that large variations in melt extent can be caused by variations in surface
reflectance characteristics. Over our entire study period, however, RACMO shows a good agreement with lake area.

## 5.4 Implications for the future

Whilst the AIS has some of the highest concentrations of surface meltwater ponding in Antarctica, the ice shelf lies in a region
that is currently thought to be largely resilient to wide scale hydrofracture (Lai et al., 2020). Substantial lateral buttressing from
the valley sides results in relatively low tensile longitudinal resistive stresses, meaning increased meltwater ponding is unlikely
to cause the rapid break-up of the ice shelf (Lai et al., 2020). However, given the vast amount of ice that is discharged through
the AIS, it is crucial that we continue to develop our understanding of how varying levels of surface meltwater can influence
hydrological and ice dynamic processes in the region. Repeated cycles of melting and re-freezing at the ice surface releases

latent heat, weakening the ice structure and making it more prone to future climatic perturbations (Hubbard et al., 2016).
Changes in temperature or precipitation patterns, in addition to predicted ocean warming, could also influence the vulnerability
of the ice shelf to melt-induced fracture. Furthermore, if meltwater starts to pond at higher elevations on a regular basis,
crevasses on steeper topography may start to undergo enhanced hydrofracture processes (Tuckett et al., 2019). The advection
of this weakened ice structure onto the ice shelf could precondition the ice shelf to further fracturing from greater volumes of

surface meltwater ponding (Dunmire et al., 2020).

The association we observe between lake area and RACMO modelled snowmelt gives us confidence in the ability of this
model to predict future melt conditions. These results show that modelled melt rates from RACMO could be used to generate
first-order predictions of surface meltwater area and volume at an annual scale for the AIS region. However, some melt
conditions that lead to the formation of lakes aren't currently well captured by RACMO, such as the influence of blue ice on

lake formation (Fig. 11). Snowmelt-albedo feedbacks have a particularly strong influence on melt rates in East Antarctica
(Jacobs et al., 2021), and further work is required to quantify this process within modelled melt estimates. Future work should
also evaluate whether a similar relationship between modelled melt and lake area occurs for other areas in Antarctica. The



surface characteristics of some regions may preclude the formation of surface lakes (e.g. if firn aquifers are present), resulting in a weaker association between modelled melt and observed lakes, even if modelled estimates are broadly accurate. It is also possible that variations in hypsometry and lateral meltwater transfer alter the lag we find between modelled melt and peak meltwater ponding (Fig. 8, Table 3).

The influence of the SAM on future meltwater cover in the study region will likely be influenced by trends in both stratospheric ozone levels and greenhouse gas emissions (Fogt & Marshall, 2020). Stratospheric ozone depletion has led to positive trends in the SAM in the austral summer season over recent decades, although there are signs that recovery of the stratospheric ozone hole is starting to counter this trend (Banerjee et al., 2020). Increases in greenhouse gas emissions have been shown to have a secondary influence on the SAM by strengthening the mid-to-high latitude temperature gradient, hence resulting in a more positive SAM (Arblaster et al., 2006). Future melt rates on the AIS will therefore likely be influenced by several competing climatic factors, with enhanced melt from regional warming and near-surface feedbacks potentially being offset by decreased melt associated with a positive SAM.

Large volumes of surface meltwater on grounded ice around the AIS (~30 % of estimated total lake area) leaves open the potential for surface to ice-bed connections to develop via hydrofracture (Krawczynski et al., 2009). Surface-melt induced variations in Antarctic ice flow have currently only been inferred to occur on northern parts of the Antarctic Peninsula (Tuckett et al., 2019). However, it is likely that surface-to-bed hydraulic connections will become more frequent as Antarctic-wide temperatures increase (Bell et al., 2017), and evidence of lake drainage events has already been identified in the grounding zone of the AIS region (Fricker et al., 2009; Pan et al., 2020; Spergel et al., 2021). The injection of surface meltwater to the ice sheet bed could also have implications on rates of ice-shelf basal melting, as a consequence of meltwater plumes emerging at the grounding line (Jacobs et al., 1992). Future work should therefore investigate the distribution and recurrence frequency of lake drainage events, and assess whether they have any impact on grounded ice flow of glaciers feeding the AIS. If such a link were found to exist, it could have significant impacts on the speed at which ice is discharged into the AIS, hence influencing rates of sea level rise.

## 6 Conclusions

We have applied an optical image band reflectance threshold-based method for identifying surface meltwater from Landsat imagery (Moussavi et al., 2020) within Google Earth Engine, enabling the automatic identification of SGLs over large spatial and temporal scales. Furthermore, our approach incorporates a robust method for assessing image visibility, allowing us to attach quantitative uncertainty estimates to mapped lake areas. By applying a time window approach and accounting for image visibility in the interpretation of results, we have generated the first continuous and consistent time series of lake area for the Amery Ice Shelf region between 2005 and 2020. We show that there is high annual variability in lake area cover in the AIS region, and that seasonal surface melt coverage is significantly influenced by variations in the SAM. Positive phases of the SAM are associated with low meltwater coverage, whilst melt seasons with a negative austral summer SAM index are typically

associated with high melt years and widespread surface melt extent. For a typical year, lake area remains low during the early melt season (November – mid December) before rapidly increasing during the second half of December. Maximum total lake area is most commonly observed during January, before sharply declining during February as lakes presumably freeze over. The spatial distribution of lakes on the ice shelf is strongly influenced by melt-albedo feedbacks, especially the exposure of blue ice from the persistent scouring of the surface by strong katabatic winds. We find a strong correlation between RACMO

modelled snowmelt and cumulative lake area, providing confidence in our ability to predict future surface meltwater ponding based on regional climate model projections in this region.

Our results demonstrate a reliable and easy to implement workflow for robustly quantifying Antarctic surface meltwater extent through time. Future work will therefore include scaling up the method to assess spatial and temporal trends in surface meltwater extent at a continent-wide scale. Such a dataset would enable a greater understanding of pan-Antarctic controls on

surface meltwater ponding, and allow us to assess how surface hydrological systems respond to varying atmospheric temperatures. This work will ultimately contribute to advancing our understanding of surface hydrological processes in Antarctica, which will have an increasingly important influence on the surface mass balance of the ice sheets in the near future.

*Data availability*. Landsat imagery is freely available from the United States Geological Survey EarthExplorer (https://earthexplorer.usgs.gov/) or via the Google Earth Engine data catalogue (https://developers.google.com/earth-engine/datasets/catalog/landsat). SAM index data, following Marshall et al. (2003), are available from the British Antarctic Survey (http://www.nerc-bas.ac.uk/public/icd/gjma/newsam.1957.2007.seas.txt). RACMO2.3p2 model data are available from the Institute for Marine and Atmospheric research Utrecht

(https://www.projects.science.uu.nl/iceclimate/models/antarctica.php). Contact: j.m.vanwessem@uu.nl

*Code availability*. The Google Earth Engine and Matlab scripts used to process the data are available at https://figshare.com/s/53b19264c7ddb5161370 (GEE) and https://figshare.com/s/ab169d57bc6d061d710b (Matlab).

*Supplement*. The supplement related to this article is available online at:

*Author contribution*. PAT, JCE, AJS, SJL and JML conceived the study. PAT developed the methodology and the GEE script, building on prior work by JML and under the supervision of JCE, AJS and SJL. AJL provided assistance developing the Matlab post-processing script. JMJ provided guidance on the climate comparison sections. JMW provided the RACMO

data and gave guidance on this section. PAT conducted all other analysis and led the manuscript writing, with input from all authors.

*Competing interests*. The authors declare that they have no conflict of interest.



*Acknowledgements.* PAT was funded by a University Post Graduate Research Committee (UPGRC) Scholarship from the University of Sheffield. JCE acknowledges a NERC independent research fellowship (grant no. NE/R014574/1). JML is supported by UKRI Future Leaders Fellowship (Grant No. MR/S017232/1). JMW acknowledges financial contributions made by the Netherlands Organization for Scientific Research (grant 866.15.201) and the Netherlands Earth System Science Center (NESSC).

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
