# Peer review of "Automated mapping of the seasonal evolution of surface meltwater and its links to climate on the Amery Ice Shelf, Antarctica"

_The Cryosphere, 2021_

## Referee Comment (RC1)

**Review of Tuckett et al. (2021)- The Cryosphere- Sammie Buzzard (Cardiff University)**

This paper describes a new, mostly automated procedure for mapping the evolution of surface melt lakes on the Amery Ice Shelf. The work uses a basis of previously published methods, combined with new tools to improve estimates and processing time, suggesting that this could be scaled up to pan-Antarctic studies.

The main innovation is through the use of Google Earth Engine for processing, and the use of an image visibility assessment to account for cloud cover, which may have caused underestimates of lake area in previous studies.

This work is a useful contribution to the field and I would be pleased to see this published. The paper is generally well presented, with some particularly well put together diagrams. However there are some issues, mainly in the clarity of the writing/ presentation that need some attention before publication.

**General comment:**

Use of RACMO data: While I absolutely appreciate that working with RACMO is a case of using the best data that's available I wasn't totally convinced by the usefulness of this part given that RACMO can't account for any lake formation/ lateral water transport and misses the influence of melt lakes on albedo, density, temperature etc. While the authors rightly state that RACMO can serve as a boundary condition (which is then limited but the accuracy of RACMO), it's compared directly with lake area here so I wasn't really sure this was a particularly informative result.

**Line by line comments:**

Line 52: Smith et al. 2020 paper not in the reference list (was hoping to look this one up, as I was unsure about the comment on the line above that most surface melt refreezes in situ and how it relates to lateral transport)

Table 1: This could probably go into the supplement as it's already published information

Figure 3: This, and Figure 2, were really well put together and really helped me to understand the process. I was a little confused by image d) however, given the size/ resolution of the image it's quite hard to distinguish between the coloured pixels (I can see black and red and maybe some blue). It may be helpful to give an indication of all the colours used, and the % of the image that each makes up?

Figure 4: This has the potential to be another really useful figure but I'm afraid I got a little lost here. Using ROI to be both the region of interest, and the % of the region of interest that's covered by the optical image is confusing. Furthermore, I wasn't sure why you were giving a value for VoI for each image, but then always dividing by 7,500- what purpose do these other VoI values serve? This could be a great visual aid with a little more clarity.

Table 2: Could these images be provided in the supplement as an example?

Section 3.4: Is all of this automated (other than the file downloading)? Could file sizes be an issue for scaling this up?

Line 259: Are these actually islands or just floating ice/ broken off ice lids? (i.e. could this be causing an underestimate in lake area?)

Line 294: How were false positives detected? What proportion of the dataset was examined for them? (Likewise for false negatives)

Line 298: I'm assuming in this paragraph you're just referring to band-thresholding techniques rather than your method as a whole?

Figure 6: Are these lakes (especially those on the floating ice shelf) as long as this image suggests, or is it a result of the same topography moving along with ice flow showing the same lake basin over several melt seasons? It would be interesting to see this image compared to e.g. the year with max lake distribution in this study to see what a single year on the ice shelf could look like.

Figure 7: Showing lake areas as a proportion of the study region area may be more useful here to spot where changes are really occurring, the low elevation changes mask anything happening as higher elevations.

Section 4.3: This section didn't seem to flow very well. I'd suggest discussing lake numbers, then lake areas, then comparing the two as this currently jumps around quite a lot.

Line 420: Is 21.0 a high or low rate? To follow on from the previous sentence it would suggest that it's not particularly high in comparison to other years, but line 415 suggests it is high. Perhaps this data could also be presented?

Line 431: Are 'accumulations of surface meltwater' lakes, or is RACMO data still included here?

Line 480: I really hate to be "that" reviewer and point you to my own work, but I think it's worth mentioning here that this has also been modelled and shown to have very important feedback for lake formation and development over multiple melt seasons

(https://agupubs.onlinelibrary.wiley.com/doi/full/10.1002/2017MS001155 for a general case, and https://tc.copernicus.org/articles/12/3565/2018/ for Larsen C specifically as with the Hubbard paper).

Line 491: There are few papers that deal specifically with foehn winds over ice shelves (e.g. https://agupubs.onlinelibrary.wiley.com/doi/10.1029/2020JD032463)

Line 546: It seems a little odd to say that the index has an effect- it's just an index. Maybe better to say they are linked/ correlated?

Line 568: Was this event in the RACMO data?

Line 600: I'd say likely rather than possible but I may be biased!

---

## Referee Comment (RC2)

**General comments:**

This manuscript presents a time series of surface meltwater area over the Amery Ice Shelf, East Antarctica, which has been automatically generated using an existing optical band threshold meltwater detection algorithm alongside a new method for calculating image visibility in Google Earth Engine. The authors use this to estimate cumulative total lake areas between 2005 and 2020 and compare this to modelled melt predictions from a regional climate model. Finally, they discuss controls on the temporal variations in surface meltwater, including the summer Southern Annular Mode index.

Although the method for surface meltwater delineation in this study is not novel, it goes beyond previous approaches (Moussavi et al., 2020) by quantifying image visibility (i.e. area of surface water missed due to cloud cover and image data coverage) with the aim of developing a consistent time series. This is a useful step forward given that most surface meltwater assessments do not explicitly quantify this. In addition, while the relationship between surface melting and large-scale climatic variability has been investigated on the Antarctic Peninsula (e.g. Bevan et al., 2018), it has so far rarely been explored as a control on meltwater on East Antarctic ice shelves. Therefore, it is my view that the findings from this manuscript are of broad interest to the cryospheric community.

In general, I would like to complement the authors on their well-written and clearly-structured manuscript. The study aims and methods are generally well explained and justified, and there is good consideration of wider controls on lake formation and variability. The results build upon previous work that has reported surface meltwater on this ice shelf by specifically accounting for image visibility in the interpretation of lake area time series. It is encouraging to see that this method can be applied to Landsat 7 imagery as well as Landsat 8.

I would like to see the authors expand their discussion around how their approach of scaling up mapped lake areas could be overpredicting lake coverage. Surface meltwater lakes tend to be highly clustered, so some pixels (regions) are much more likely to be covered by surface meltwater than others. This could produce potentially large lake area overestimates, and if applied over other regions of Antarctica could affect the accuracy of surface meltwater time series.

Secondly, although the authors discuss misclassification errors in Section 4.1, I think this could expanded, including how these were verified. I question whether this one study area alone is sufficient to be able to determine the applicability of this approach on a pan-ice sheet scale. Misclassifications could be a lot higher across other ice shelves (especially in dirty ice/high debris cover/topographically-complex regions). I think this should be acknowledged more in the text when discussing transferability across other ice shelves.

Lastly, the abstract is quite a bit lengthier than this journal's suggested limit of 150-250 words and could provide a more concise summary.

Once the authors address this point and my specific comments below, I can therefore recommend that this manuscript is suitable for publication in The Cryosphere.

**Specific comments:**

Line 37: widespread  → 'around'  (as meltwater is restricted to the ice-sheet margins)

Line 42: I find 'concentrated at the break of slope' an odd phrase to describe katabatic winds, as they descend the escarpment region from the interior – suggest rewording.

Line 57:  the flow of grounded ice → accelerate?

Line 109: I question whether this one study area alone is sufficient to be able to determine the applicability of this approach on a pan-ice sheet scale. Although it is a large ice shelf with large numbers of lakes, it is a relatively 'uncomplicated' area – the performance could be different in regions of dirty ice/with high debris cover or wind-blown sediment, for example the McMurdo Ice Shelf. I don't think you can be completely confident about its continental applications until at least multiple test sites have been trialled.

Line 115: (Figure 1) – Could you use a more up-to-date basemap and coastline in this figure, for example from the Antarctic Digital Database which has a high resolution coastline updated since the 2019 'loose tooth' calving event (https://data.bas.ac.uk/collections/e74543c0-4c4e-4b41-aa33-5bb2f67df389/)?

Line 158: specify this is the NDWI adapted for ice (red and blue bands), typically written NDWI$_{ice}$.

Line 163: It isn't immediately clear what period a time window covers - is this variable, and are there always two time windows per month?

Line 215: I found this Figure quite difficult to get my head around, especially VoI vs TotV – could it be clarified somehow?

Line 259: I suggest maybe clarifying that an 'island' refers to an ice lid within the lake centre.

Line 266: Why was this Depoorter et al. (2013) grounding line dataset used rather than, for example, the MEaSUREs dataset (Rignot et al., 2016) derived from DinSAR measurements from 1992-2014?

Line 274: Please specify ERA-interim resolution.

Line 275: Is the box covering the study region that is referred to here the same red box shown in Figure 1, or a different extent? How were mean monthly summed melt values extracted from within this box – did you include values from any RACMO grid cells that intersected this box?

Line 285: temperatures, and therefore surface melting?

Line 293: which part of the ice shelf/grounded ice were false positives generally located?

Line 295: how did you estimate lake depth?

Line 300: (Figure 5) Consider adding insets to show locations on ice shelf.

Line 355: (Figure 7) Colours are difficult to distinguish (e.g. between December and January lakes on Panel a). Could you also indicate the maximum elevation of the study region?

Line 363: 'The highest <cumulative> number of lakes were observed' because there are higher lake numbers in individual time windows in 2014, 2018 and 2019.

Line 365: I would refer the reader to the relevant sub-panels here, e.g. Fig 8b. Similarly, Fig 8a-b on line 373.

Line 392: I find it quite difficult to see the seasonality discussed in the text given the bars are quite close together – maybe you could choose one melt season to do an additional seasonal plot? Also a question regarding numbers of lakes in Panel (a) – how did you account for the Scan Line Corrector striping 'dissecting' larger lakes into multiple polygons, and could this potentially have affected  lake totals presented pre-Jan 13?

Line 406: 2013/2014 modelled melt total is higher than in 2016/17, so this doesn't appear to be correct?

Line 409: Again, I find it hard to pick out these minor interannual variations in modelled melt rates using the axis scale you have chosen.. for example, distinguishing whether melt peaks in December or January in different years. Also, wondering why you have chosen to present modelled melt in units kg m$^2$ rather than the typically-used mm w.e. day$^{-1}$?

Line 453: 'whereby the temporal resolution can be varied' – please clarify.

Line 555: I agree that Figure 11 shows surface scouring and blue ice exposure, likely generated by katabatic winds, but can you really say that this was enhanced katabatic wind scouring compared to other melt years? Is there evidence of this in the satellite imagery for other melt seasons?

Line 571: '[…] does not specifically include a representation of blue ice', and because it is difficult for regional climate models like RACMO to resolve these melt feedbacks at this scale (27 km).

Line 589: Good point.

Line 593: Your analysis focuses on lake area, so I suggest you remove the reference to volume here.

**Technical corrections:**

Line 51: italicize 'in situ'

Line 53: 'ice shelf flexure'

Line 156: hyphenate 'mask based'

Line 265: hyphenate 'meltwater filled'

Line 281: delete 'the' large-scale

Line 296: hyphenate 'sediment laden'

Line 481: hasn't → has not

Line 504: Roi Bedouin → Roi Baudouin

Line 573: 'under predicts' → one word

Line 594: aren't → are not

**References**

Bevan, S. L., Luckman, A.J., Kuipers Munneke, P., Hubbard, B., Kulessa, B., & Ashmore, D. W. (2018). Decline in surface melt duration on Larsen C Ice Shelf revealed by the advanced scatterometer (ASCAT). Earth and Space Science, 5. https://doi.org/10.1029/2018EA000421.

Rignot, E., J. Mouginot, and B. Scheuchl. (2016) MEaSUREs Antarctic Grounding Line from Differential Satellite Radar Interferometry, Version 2. [Indicate subset used]. Boulder, Colorado USA. NASA National Snow and Ice Data Center Distributed Active Archive Center. doi: https://doi.org/10.5067/IKBWW4RYHF1Q.

---

## Author Comment (AC2)

**Response to Reviewer Comments**

**Reviewer #1 (Sammie Buzzard, Cardiff University)**

This paper describes a new, mostly automated procedure for mapping the evolution of surface melt lakes on the Amery Ice Shelf. The work uses a basis of previously published methods, combined with new tools to improve estimates and processing time, suggesting that this could be scaled up to pan-Antarctic studies. The main innovation is through the use of Google Earth Engine for processing, and the use of an image visibility assessment to account for cloud cover, which may have caused underestimates of lake area in previous studies. This work is a useful contribution to the field and I would be pleased to see this published. The paper is generally well presented, with some particularly well put together diagrams. However there are some issues, mainly in the clarity of the writing/ presentation that need some attention before publication.

We would like to thank reviewer 1 for a useful and constructive review. We are glad that the reviewer deems our work a 'useful contribution to the field', and are pleased that our figures were helpful for interpreting the methodology. To further aid the clarity of the manuscript, we have considered all and adopted many of the changes recommended by reviewer #1, including altering selected figures and expanding a few sections of the discussion.

Use of RACMO data: While I absolutely appreciate that working with RACMO is a case of using the best data that's available I wasn't totally convinced by the usefulness of this part given that RACMO can't account for any lake formation/ lateral water transport and misses the influence of melt lakes on albedo, density, temperature etc. While the authors rightly state that RACMO can serve as a boundary condition (which is then limited but the accuracy of RACMO), it's compared directly with lake area here so I wasn't really sure this was a particularly informative result.

We appreciate that a comparison between RACMO and lake area has many limitations (as already stated in text), and we have altered some of the wording to make this point further. However, we believe that despite the limitations, this upper bound for meltwater availability provides a useful first-order estimate of meltwater production, and hence likelihood of meltwater ponding. Indeed, it is especially interesting that we observe such a clear correlation between the two variables in light of all the caveats, indicating that at broad spatial (catchment) and temporal scales (cumulative annual melt and lake totals in Figure 9), total lake area is strongly influenced by the amount of melt generated. The implication is that most meltwater produced in a catchment enters lakes/channels observable at the resolution of Landsat-7/8, even if the distribution and evolution of water storage/ transport is likely to vary over shorter temporal and spatial scales. We therefore believe that providing a consistency test between the two variables provides a valuable element to the results, especially given there is such high variability in meltwater extent between melt seasons.

**Line by line comments:**

Line 52: Smith et al. 2020 paper not in the reference list (was hoping to look this one up, as I was unsure about the comment on the line above that most surface melt refreezes in situ and how it relates to lateral transport).

Reference added.

Table 1: This could probably go into the supplement as it's already published information

Table moved to supplement.

Figure 3: This, and Figure 2, were really well put together and really helped me to understand the process. I was a little confused by image d) however, given the size/ resolution of the image it's quite hard to distinguish between the coloured pixels (I can see black and red and maybe some blue). It may be helpful to give an indication of all the colours used, and the % of the image that each makes up?

We have added a new image for (d), which makes it easier to identify lakes, along with a clearer colour scale. We have also added a key showing which image each colour represents, and the % contribution from each image. Furthermore, this section of the figure is now replicated in the Supplement as part of a new figure, which further explains the process (following reviewer #1's comment regarding Table 2).

Figure 4: This has the potential to be another really useful figure but I'm afraid I got a little lost here. Using ROI to be both the region of interest, and the % of the region of interest that's covered by the optical image is confusing. Furthermore, I wasn't sure why you were giving a value for VoI for each image, but then always dividing by 7,500- what purpose do these other VoI values serve? This could be a great visual aid with a little more clarity.

We agree that using ROI when referring to different things was confusing. The term used in the figure has been changed to 'ROI Coverage'. Furthermore, TotV has now been removed for simplicity, and VoI is written in full to avoid confusion. The figure caption has been adjusted accordingly, and now provides more explanation of why 'Visible over Ice' scores are divided by 7,500 (the clear-sky ice mask value in this example).

Table 2: Could these images be provided in the supplement as an example?

We have added a new figure to the Supplement (Figure S5), containing the six images used in this example.

Section 3.4: Is all of this automated (other than the file downloading)? Could file sizes be an issue for scaling this up?

We have added a new sentence at the end of the section, confirming that all post-processing steps are automated in Matlab, and giving an indication of the time taken. File sizes are dependent on the number of lakes mapped. Since Amery is the region with the largest number/area of lakes in Antarctica, we are confident that scaling up will not be an issue in terms of run time/file sizes.

Line 259: Are these actually islands or just floating ice/ broken off ice lids? (i.e. could this be causing an underestimate in lake area?)

We have clarified that 'island' typically refers to an ice lid.

Line 294: How were false positives detected? What proportion of the dataset was examined for them? (Likewise for false negatives)

In our manuscript we use the lake detection thresholds of Moussavi et al. (2020). They compared automatically mapped lake areas against manually-digitized lake polygons, in addition to visually comparing mapped lakes against ~1000 satellite image tiles. They found that the thresholds applied produced consistent and reliable lake results, with overall accuracies of >95% from Landsat-8 images. As the thresholds had not been applied to Landsat-7 before, we carried out an additional comparison between outputs from Landsat-7 and Landsat-8 images taken over the same region ~24 hrs apart (see Supplementary section). This revealed very similar results (see S4), with variations typically associated with the margins of lakes/rivers, despite the striping in the Landsat-7 imagery. In

addition to this, we manually checked mapped lakes against satellite imagery for approximately ~10% of randomly selected time windows, to provide a broad check of the method's performance. We have added two sentences on line 285 to clarify this.

Line 298: I'm assuming in this paragraph you're just referring to band-thresholding techniques rather than your method as a whole?

Correct. We have altered our wording to clarify this.

Figure 6: Are these lakes (especially those on the floating ice shelf) as long as this image suggests, or is it a result of the same topography moving along with ice flow showing the same lake basin over several melt seasons? It would be interesting to see this image compared to e.g. the year with max lake distribution in this study to see what a single year on the ice shelf could look like.

We have added two new figures to the Supplement (Figures S6 and S7), showing lake distribution in years with minimum (2010/11) and maximum (2005/06) total lake area. Comparison of these additional figures with Figure 6 shows that the long lakes in question were present during the 2005/06 melt season only, and were not the result of lake basins moving with ice flow over multiple years.

Figure 7: Showing lake areas as a proportion of the study region area may be more useful here to spot where changes are really occurring, the low elevation changes mask anything happening as higher elevations.

We initially used lake area proportions (as suggested) when making this figure. However, since the low elevation lake areas are so much greater, it did not offer a particular improvement in distinguishing lake area differences at higher elevations. Furthermore, differences in lake areas throughout a melt season were more difficult to observe using this approach (the monthly lines in (a) were almost identical when using lake proportions). We therefore decided that it was more important to show monthly trends using absolute values of lake area throughout a melt season. The use of absolute values also enables better comparison between a high (a) and low (b) melt year. We have therefore retained the original figure.

Section 4.3: This section didn't seem to flow very well. I'd suggest discussing lake numbers, then lake areas, then comparing the two as this currently jumps around quite a lot.

This section has been re-ordered, as per reviewer #1's suggestion.

Line 420: Is 21.0 a high or low rate? To follow on from the previous sentence it would suggest that it's not particularly high in comparison to other years, but line 415 suggests it is high. Perhaps this data could also be presented?

We thank reviewer #1 for highlighting this mistake. 'Relatively high melt' has been changed to 'relatively low melt'. These data are presented in Figure 8c, and are also now shown in Supplement Figure S8, making it easier to view monthly changes in RACMO snowmelt estimates during each melt season.

Line 431: Are 'accumulations of surface meltwater' lakes, or is RACMO data still included here?

This is referring to lake areas and not RACMO. This has now been clarified.

Line 480: I really hate to be "that" reviewer and point you to my own work, but I think it's worth mentioning here that this has also been modelled and shown to have very important feedback for lake formation and development over multiple melt seasons

(https://agupubs.onlinelibrary.wiley.com/doi/full/10.1002/2017MS001155 for a general case, and https://tc.copernicus.org/articles/12/3565/2018/ for Larsen C specifically as with the Hubbard paper).

We have added a sentence on the importance of ice slabs on lake development, based on modelling studies (including the suggested reference).

Line 491: There are few papers that deal specifically with foehn winds over ice shelves (e.g. https://agupubs.onlinelibrary.wiley.com/doi/10.1029/2020JD032463)

Reference added.

Line 546: It seems a little odd to say that the index has an effect- it's just an index. Maybe better to say they are linked/ correlated?

Changed to 'linked'.

Line 568: Was this event in the RACMO data?

The RACMO data provides monthly summed melt values, meaning it is impossible to separate out transient increases in melt (e.g. from a katabatic event) from continuous melt over the month. There is an increase in melt in December 2014, which is displayed in Figure 8 and Supplementary Figure S8.

Line 600: I'd say likely rather than possible but I may be biased!

Changed

**Reviewer #2 (Anonymous)**

This manuscript presents a time series of surface meltwater area over the Amery Ice Shelf, East Antarctica, which has been automatically generated using an existing optical band threshold meltwater detection algorithm alongside a new method for calculating image visibility in Google Earth Engine. The authors use this to estimate cumulative total lake areas between 2005 and 2020 and compare this to modelled melt predictions from a regional climate model. Finally, they discuss controls on the temporal variations in surface meltwater, including the summer Southern Annular Mode index.

Although the method for surface meltwater delineation in this study is not novel, it goes beyond previous approaches (Moussavi et al., 2020) by quantifying image visibility (i.e. area of surface water missed due to cloud cover and image data coverage) with the aim of developing a consistent time series. This is a useful step forward given that most surface meltwater assessments do not explicitly quantify this. In addition, while the relationship between surface melting and large-scale climatic variability has been investigated on the Antarctic Peninsula (e.g. Bevan et al., 2018), it has so far rarely been explored as a control on meltwater on East Antarctic ice shelves. Therefore, it is my view that the findings from this manuscript are of broad interest to the cryospheric community.

In general, I would like to complement the authors on their well-written and clearly-structured manuscript. The study aims and methods are generally well explained and justified, and there is good consideration of wider controls on lake formation and variability. The results build upon previous work that has reported surface meltwater on this ice shelf by specifically accounting for image visibility in the interpretation of lake area time series. It is encouraging to see that this method can be applied to Landsat 7 imagery as well as Landsat 8.

We would like to thank reviewer #2 for their comments, and are pleased that the reviewer acknowledges the inclusion of image visibility assessments as an important step forward for producing lake area time series. We have considered all the points raised by reviewer #2 and altered the manuscript accordingly.

I would like to see the authors expand their discussion around how their approach of scaling up mapped lake areas could be overpredicting lake coverage. Surface meltwater lakes tend to be highly clustered, so some pixels (regions) are much more likely to be covered by surface meltwater than others. This could produce potentially large lake area overestimates, and if applied over other regions of Antarctica could affect the accuracy of surface meltwater time series.

Reviewer #2 is correct that in regions where lakes are highly clustered within an ROI, our maximum estimates are prone to error, because our upper lake area estimate assumes an even distribution of lake pixels across each ROI. We do not, however, believe that this will result in major or systematic overestimates. An ROI will typically be covered by several images within a time window, meaning it is highly unlikely that all images over our ~2 weekly sampling period either only cover a clustered (or dispersed) section of lakes, or that there is rapid change in lake distribution within a time window. Thus, sometimes our approach may overestimate lake areas slightly (if cloud cover is focused over clusters of lakes), and sometimes underestimate them (if cloud cover is focused over lake free areas), but we expect these effects to balance out over multi-annual timescales. These points are now made in the manuscript at the end of Section 5.1. Although it won't be linear, any over/under-estimation is likely to become more significant as the visibility score decreases. For this study visibility scores were typically on the order of 70-90%, and so any uncertainty associated with this is likely to be small. We acknowledge, however, that for cloudier regions of the ice sheet this could

become more significant or reliant on a single image. In this situation, it may be appropriate to increase the temporal window, and would be an interesting future avenue to explore. We note that this is the first attempt to quantify the uncertainty induced by cloud cover, and future improvements may adapt our technique.

Secondly, although the authors discuss misclassification errors in Section 4.1, I think this could expanded, including how these were verified. I question whether this one study area alone is sufficient to be able to determine the applicability of this approach on a pan-ice sheet scale. Misclassifications could be a lot higher across other ice shelves (especially in dirty ice/high debris cover/topographically-complex regions). I think this should be acknowledged more in the text when discussing transferability across other ice shelves.

We thank reviewer #2 for raising this point, and agree that it is useful to provide more discussion on scaling up the method for continent-wide mapping. We have clarified that Moussavi et al. (2020) established thresholds (which we apply) based on spectral analysis of four different ice shelves around Antarctica, covering a range of surface conditions and melt characteristics. Based on their conclusions, we believe that their (and thus also our) thresholds are suitable for continent-wide use. However, we appreciate that there will still be regions of Antarctica (such as the McMurdo ice shelf, which wasn't used in the Moussavi et al. (2020) study) where the thresholds may need adjusting to avoid misclassification errors. This is now acknowledged on Line 460, with specific reference to ice shelves with large areas of dirty ice or surface debris. We have additionally clarified what verification steps were taken in Section 4.1 – see comment response to line 294 from reviewer #1.

Lastly, the abstract is quite a bit lengthier than this journal's suggested limit of 150-250 words and could provide a more concise summary. Once the authors address this point and my specific comments below, I can therefore recommend that this manuscript is suitable for publication in The Cryosphere.

The abstract has been significantly shortened to approximately 250 words, and now offers a more concise summary of the paper.

**Line by line comments:**

Line 37: widespread across → 'around' (as meltwater is restricted to the ice-sheet margins)

Changed.

Line 42: I find 'concentrated at the break of slope' an odd phrase to describe katabatic winds, as they descend the escarpment region from the interior – suggest rewording.

Rephrased.

Line 57: affect the flow of grounded ice → accelerate?

The injection of surface water to the bed can cause both accelerations and decelerations in ice velocity, due to changes in subglacial water and effective pressure. We therefore retain the original wording.

Line 109: I question whether this one study area alone is sufficient to be able to determine the applicability of this approach on a pan-ice sheet scale. Although it is a large ice shelf with large numbers of lakes, it is a relatively 'uncomplicated' area – the performance could be different in regions of dirty ice/with high debris cover or wind-blown sediment, for example the McMurdo Ice

Shelf. I don't think you can be completely confident about its continental applications until at least multiple test sites have been trialled.

We have altered the wording to specify that, in this section, we are assessing the suitability of the method from a computational perspective, rather than the suitability of the threshold-based method itself. We have also added a section to the discussion (Line 460) where we discuss the applicability of the method at a continent-wide scale in terms of the thresholds used, including possible differences in the spectral properties of ice shelves. See response to Reviewer #2's general comment, where we discuss this further.

Line 115: (Figure 1) – Could you use a more up-to-date basemap and coastline in this figure, for example from the Antarctic Digital Database which has a high resolution coastline updated since the 2019 'loose tooth' calving event (https://data.bas.ac.uk/collections/e74543c0-4c4e-4b41-aa33-5bb2f67df389/)?

We thank Reviewer #2 for suggesting a more recent dataset. We have replaced the Depoorter (2013) dataset with the suggested coastline dataset from the Antarctic Digital Database, and have replaced the basemap with the Landsat Image Mosaic of Antarctica. The figure caption has been adjusted accordingly.

Line 158: specify this is the NDWI adapted for ice (red and blue bands), typically written NDWIice.

Done.

Line 163: It isn't immediately clear what period a time window covers - is this variable, and are there always two time windows per month?

We have added a sentence on line 121 (when time windows were first introduced) explaining that for this study, the' time window' input was set as two time-windows per month. This was chosen instead of a set number of days for ease of comparison between years. The time window input in Figure 2 has been changed to 'Time window length' to help avoid confusion.

Line 215: I found this Figure quite difficult to get my head around, especially VoI vs TotV – could it be clarified somehow?

The description of this figure has been changed to add more clarity, following comments from both reviewers. The acronyms used in the figure have also been altered, including the removal of 'TotV' as this was misleading.

Line 259: I suggest maybe clarifying that an 'island' refers to an ice lid within the lake centre.

Done.

Line 266: Why was this Depoorter et al. (2013) grounding line dataset used rather than, for example, the MEaSUREs dataset (Rignot et al., 2016) derived from DinSAR measurements from 1992-2014?

Whilst Reviewer #2 is correct that more recent grounding line datasets have been published (such as Rignot et al., 2016), these datasets are less complete in terms of continuous coverage of the study region. The Depoorter et al. (2013) dataset is the most recent, continuous open-access dataset that we were able to find. Given that the grounding line location is not central to the study, and was used simply as a broad measure of quantifying lake areas on/off grounded ice, we deemed this dataset to be more appropriate.

Line 274: Please specify ERA-interim resolution.

Done.

Line 275: Is the box covering the study region that is referred to here the same red box shown in Figure 1, or a different extent? How were mean monthly summed melt values extracted from within this box – did you include values from any RACMO grid cells that intersected this box?

RACMO values were extracted from an area covering the entire study region, which matched the red boxes (lake area ROIs) as closely as possible whilst lining up with the pixel spacing of the RACMO grid. We have changed the wording to clarify that extracted RACMO pixel values were then summed across the study region.

Line 285: temperatures, and therefore surface melting?

Added.

Line 293: which part of the ice shelf/grounded ice were false positives generally located?

Sentence added stating that there was no particular spatial clustering of false positives.

Line 295: how did you estimate lake depth?

Lake depths were not calculated as part of this study. This sentence has been re-phrased to clarify that lakes 'appeared to be very deep' due to their dark colour.

Line 300: (Figure 5) Consider adding insets to show locations on ice shelf.

We tried this, but insets covered over important parts of the figure and could not be added easily without overcrowding the images. We have added two coloured stars to Figure 1 indicating the location of the images shown in Figure 5. The figure caption also describes the location of each region in relation to two prominent Massifs, which are both visible and labelled on Figure 1.

Line 355: (Figure 7) Colours are difficult to distinguish (e.g. between December and January lakes on Panel a). Could you also indicate the maximum elevation of the study region?

The colour scale has been changed to make it easier to distinguish between months. The highest elevation band has been changed from '>1000', to '1000-2517' to indicate the maximum elevation within the study area.

Line 363: 'The highest <cumulative> number of lakes were observed' because there are higher lake numbers in individual time windows in 2014, 2018 and 2019.

Added 'cumulative'.

Line 365: I would refer the reader to the relevant sub-panels here, e.g. Fig 8b. Similarly, Fig 8a-b on line 373.

Done.

Line 392: I find it quite difficult to see the seasonality discussed in the text given the bars are quite close together – maybe you could choose one melt season to do an additional seasonal plot? Also a question regarding numbers of lakes in Panel (a) – how did you account for the Scan Line Corrector striping 'dissecting' larger lakes into multiple polygons, and could this potentially have affected the lake totals presented pre-Jan 13?

We appreciate that due to the length of the time series, it is difficult to observe seasonal changes in lake area in some melt seasons. We have therefore added a new figure to the Supplement (Figure

S8), displaying a zoomed in version of the lake area time series for each melt season. Each yearly panel also displays RACMO melt, making it easier to observe interannual variations in modelled melt rates.

Reviewer #2 is correct that the SLC striping (influencing Landsat 7 images before 2013 included in our study) will have dissected some large lakes, hence resulting in slight overestimates in lake numbers. However, the spacing of the SLC stipes, the average size of lakes, and the scale of lake numbers involved, means that such overestimates will have been negligible. The vast majority of lakes are too small to have been influenced, meaning only the largest, longest lakes will have been affected. We therefore deemed it unnecessary to try to account for this in lake number totals, especially given that the main focus of analysis is on lake areas, which does account for SLC striping through the use of visibility assessments.

Line 406: 2013/2014 modelled melt total is higher than in 2016/17, so this doesn't appear to be correct?

This is incorrect. As shown in Figure 9, cumulative snowmelt in 2016/17 was slightly higher than in 2013/14. The spike in Figure 8c is higher in 2013/14 due to a single high month of melt, but 2016/17 had moderate levels of melt sustained over a longer time period (indicated by a wider spike in Figure 8c), resulting in a higher overall total. These differences in RACMO melt can now be viewed more clearly in Supplement Figure S8.

Line 409: Again, I find it hard to pick out these minor interannual variations in modelled melt rates using the axis scale you have chosen. for example, distinguishing whether melt peaks in December or January in different years. Also, wondering why you have chosen to present modelled melt in units kg m2 rather than the typically-used mm w.e. day-1?

See response to Line 392. Interannual variations in modelled melt can now be viewed more clearly in Supplement Figure S8. The units have been changed from kg m$^2$ to mm w.e.

Line 453: 'whereby the temporal resolution can be varied' – please clarify.

We have clarified that this is referring to the length of time-windows (e.g. daily, monthly or yearly mapping).

Line 555: I agree that Figure 11 shows surface scouring and blue ice exposure, likely generated by katabatic winds, but can you really say that this was enhanced katabatic wind scouring compared to other melt years? Is there evidence of this in the satellite imagery for other melt seasons?

Whilst surface scouring does occur on several years, 2014/15 stood out as a particularly clear example when compared to satellite images from other melt seasons, and as an outlier in Figure 9. Furthermore, other years with high levels of wind scouring had higher predicted RACMO melt rates, making it harder to separate out the two influences. In this example (Figure 11), we can be confident that pre-conditioning of the ice surface from wind scouring is the primary reason why there is such high meltwater ponding, given the low RACMO melt estimate.

Line 571: '[…] does not specifically include a representation of blue ice', and because it is difficult for regional climate models like RACMO to resolve these melt feedbacks at this scale (27 km).

Added.

Line 589: Good point.

Line 593: Your analysis focuses on lake area, so I suggest you remove the reference to volume here.

Agreed, reference to volumes has been removed.

**Technical corrections:**

Line 51: italicize 'in situ' – The Cryosphere submission guidelines states not to italicize common Latin phrases, such as in situ.

Line 53: 'ice shelf flexure' – Changed.

Line 156: hyphenate 'mask based' – Disagree, this should not be hyphenated.

Line 265: hyphenate 'meltwater filled' – Changed.

Line 281: delete 'the' large-scale – Changed.

Line 296: hyphenate 'sediment laden' – Changed.

Line 481: hasn't → has not – Changed.

Line 504: Roi Bedouin → Roi Baudouin – Changed.

Line 573: 'under predicts' → one word – Changed.

Line 594: aren't → are not – Changed.